# Exploring the trade-offs between electric heating policy and carbon mitigation in China

Jianxiao Wang [1,2], Haiwang Zhong [1✉], Zhifang Yang[3], Mu Wang[1], Daniel M. Kammen [4✉], Zhu Liu[5], Ziming Ma[1], Qing Xia[1] & Chongqing Kang[1]

China has enacted a series of policies since 2015 to substitute electricity for in-home combustion for rural residential heating. The Electric Heating Policy (EHP) has contributed to significant improvements in air quality, benefiting hundreds of millions of people. This shift, however, has resulted in a sharp increase in electric loads and associated carbon emissions. Here, we show that China's EHP will greatly increase carbon emissions. We develop a theoretical model to quantify the carbon emissions from power generation and rural residential heating sectors. We found that in 2015, an additional 101.69–162.89 megatons of $CO_2$ could potentially be emitted if EHP was implemented in 45–55% of rural residents in Northern China. In 2020, the incremental carbon emission is expected to reach 130.03–197.87 megatons. Fortunately, the growth of carbon emission will slow down due to China's urbanization progress. In 2030, the carbon emission increase induced by EHP will drop to 119.19–177.47 megatons. Finally, we conclude two kinds of practical pathways toward low-carbon electric heating, and provide techno-economic analyses.

[1] State Key Laboratory of Control and Simulation of Power Systems and Generation Equipment, Department of Electrical Engineering, Tsinghua University, 100084 Beijing, China. [2] State Key Laboratory of Alternate Electrical Power System with Renewable Energy Sources, School of Electrical and Electronic Engineering, North China Electric Power University, 102206 Beijing, China. [3] State Key Laboratory of Power Transmission Equipment & System Security and New Technology, College of Electrical Engineering, Chongqing University, 400030 Chongqing, China. [4] Energy and Resources Group, and Goldman School of Public Policy, University of California, Berkeley, CA 94720, USA. [5] Department of Earth System Science, Tsinghua University, 100084 Beijing, China. ✉email: zhonghw@tsinghua.edu.cn; kammen@berkeley.edu

Chhina has experienced continuous and dramatic development of the economy and industry over the past three decades[1]. However, as the world's largest coal consumer and coal-derived electricity producer, one consequence of the resulting massive consumption of fossil fuels is the rise of emerging greenhouse gas and air pollution emissions[2,3], posing serious threats to global warming and human health[4].

During the "13th Five-Year Plan" in China, it has become a national strategy to develop a clean-energy society and to preserve the ecological environment[5]. On the one hand, as a promise to the world, China has set an ambitious target to limit the national carbon footprint. In 2015, China agreed on the Paris Agreement and declared that the carbon emissions per GDP in 2030 must decrease by 60–65% of the value in 2005[6]. On the other hand, a wide variety of domestic actions have been taken to mitigate carbon dioxide and air pollution, e.g., improving energy efficiency[7], facilitating renewable and sustainable energy[8], enhancing forest carbon sequestration[9], etc.

At the same time, one of the main sources of the air pollutants in Northern China is rural residents' burning raw coal for heating. Due to the relatively low price and high heat value, raw coal has long been a primary heating resource in Northern China in winter. However, without desulfurization and denitrification, the $SO_2$, $NO_x$ and other air pollutants from in-home combustion are directly emitted in the atmosphere, thereby resulting in severe environmental pollution[10]. In the Beijing–Tianjin–Hebei region, the annual rural raw coal consumption generally reaches over 40 million tons, contributing to approximately 15% of $SO_2$, 4% of $NO_x$ and 23% of particles, respectively[11]. Realistic evidence in Northern China shows that the air quality in winter usually gets much worse than that in summer (Supplementary Fig. 1)[12].

Recent years have witnessed the Chinese government's great efforts to reduce the carbon and pollutant emissions from rural residents. A series of Electric Heating Policies (EHPs) has been issued since 2015, which enforces strict regulations to substitute electric heating in place of raw coal in Northern China. For example, in April 2015, the "Action plan for the clean and efficient coal" issued by the National Energy Administration declared that the use of coal with over 16% ash or 1% sulfur content is prohibited[13]. Another policy, "Instructions for substituting electric heating for coal", issued by the National Development and Reform Commission in May 2016, showed a goal of reducing 130 megatons of coal for rural heating from 2016 to 2020 in China[14]. Consequently, many provinces such as Hebei and Shanxi have issued regional action plans to popularize electric heating for raw coal abatement.

Up till now, China's EHP has contributed to significant improvements in air quality and carbon dioxide reductions from rural residential sectors. However, this shift has resulted in a sharp increase in electric loads and may even lead to a higher level of carbon emissions from power generation. Empirical evidence shows that in January 2018, the State Grid Corporation of China (SGCC) encountered a dramatic electric load increase caused by the EHP and required more electricity from coal-fired power plants. Compared with 2017, the largest daily electricity consumption in January 2018 increased by over 15%[15]. Therefore, the conflict between China's EHP and national carbon mitigation has been exposed with the rapid development of electric heating.

A wide variety of existing literature has investigated the environmental impacts of China's residential heating sectors, including the estimation for carbon and pollutant emissions[16–20], the policy making and analysis for emission control[21–24], and the influence on life expectancy and human health[25–28]. Yet few studies have quantified the greenhouse impacts caused by electric heating in China or explored the emerging incompatibility between China's EHP and carbon mitigation. Therefore, we aim

to quantify the extent that China's EHP can contribute to national carbon emissions in this paper. To quantify $CO_2$ induced by China's EHP, we propose a theoretical model considering both power generation and rural residential heating sectors. We explore the link between China's EHP and national carbon mitigation, and analyze the key factors leading to the diverse performance of the policy implementation in different regions. To address the incompatibility, we provide policy suggestions for China and other countries with similar situations to facilitate the accommodation of renewable energy and to improve electric heating efficiency.

## Results

**Provincial carbon emissions caused by Electric Heating Policy.** We quantify the carbon emissions from power generation and rural residential heating in four provinces in Northern China during the heating season in 2015. According to the data in 11 cities in Northern China (Supplementary Fig. 2 and Supplementary Table 1), electric heaters (EHs) generally account for 79.24–100% of the electric heating devices among rural residents. Instead, some residents use heat pumps (HPs) and photovoltaic-powered electric heating (PVEH). The average proportions of EHs, HPs and PVEH are 91.01%, 6.58%, and 2.41%, respectively. As illustrated in Fig. 1, the implementation of EHP leads to a significant increase in the provincial carbon emissions. Here we focus on two major sources of uncertainties, i.e., policy implementation rate (PIR) and electric heating mix (EHM). PIR refers to the population of rural residents using electric heating over that of provincial rural residents. "Plan for Winter Clean Heating in Northern China (2017–2021)"[29], issued by National Development and Reform Commission in 2017, requires 50% of rural residents in Beijing, Tianjin and 14 other provinces in Northern China to substitute electric heating for raw coal by 2019. We design three comparative cases with the PIR equaling 45%, 50%, and 55%, respectively. EHM refers to the proportions of EHs, HPs and PVEH. The proportions of EHs, HPs and PVEH are

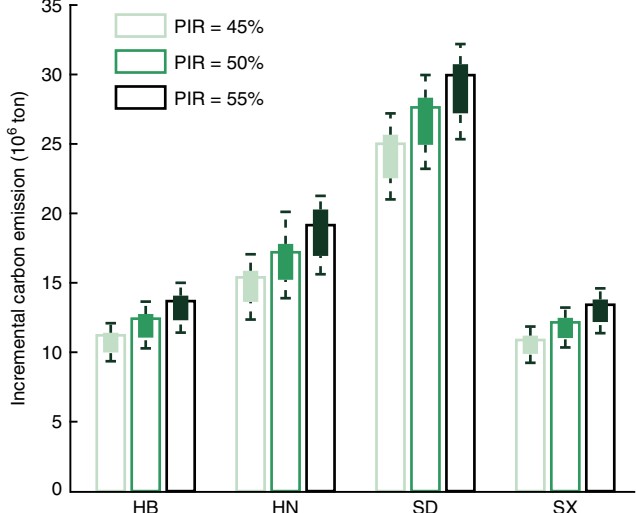

**Fig. 1 Carbon emission estimation in Hebei (HB), Henan (HN), Shandong (SD), and Shanxi (SX) provinces after implementing Electric Heating Policy in 2015.** Each three bars for a province represent the incremental carbon emissions with an average electric heating mix when the policy implementation rate (PIR) equals 45%, 50%, and 55%, respectively. Each box represents the incremental carbon emission acquired by scanning the electric heating mix ($n = 12$). The minimum/maximum of each box indicates the minimal/maximal value of incremental carbon emission, and the lower and upper percentiles are 25% and 75%, respectively.

scanned within the intervals [80%, 100%], [0, 20%], and [0, 10%], respectively.

Considering the joint uncertainties, the incremental carbon emissions in Hebei (HB), Henan (HN), Shandong (SD), and Shanxi (SX) reach [9.32, 14.97], [12.32, 21.23], [20.97, 32.16], and [9.21, 14.57] megatons, respectively. Base case is designed with 50% PIR and the average EHM. In the base case, SD releases the largest amount of $CO_2$ equaling 27.63 megatons, while the other three provinces, i.e., HB, HN, and SX, produce 12.42, 17.20, and 12.15 megatons $CO_2$, respectively.

We observe that the uncertainty in EHM (the boxplots) may yield a greater impact on the incremental carbon emissions than that in PIR (the bars). Based on the average EHM, the largest deviations of carbon emission induced by PIR uncertainty are 2.46, 3.76, 4.94, and 2.54 megatons in HB, HN, SD, and SX, respectively. However, on the premise of a fixed PIR, the largest deviations of carbon emission caused by EHM uncertainty can reach 3.58, 6.22, 6.85, and 3.22 megatons in the four provinces, respectively.

Our results demonstrate that the diversity of provincial carbon emissions mainly comes from three key factors: (i) the climate conditions such as ambient air temperature (AAT), (ii) the rural resident population (RRP) using electric heating and iii) the thermal coal consumption rate (TCCR). The AAT has a direct impact on household coal consumption and electric heating load (Fig. 2a). The lower the AAT is, the more heat energy is needed to maintain the indoor temperature. In Fig. 2a, the median values of hourly AAT in HN and SD in winter are 5.45 and −0.56 °C, respectively. As a result, SD has the highest daily average electric heating load of a single household among the four provinces, equaling 54.35 kWh, while the value in HN is the lowest, equaling 36.71 kWh. Figure 2a also shows that the incremental electric heating loads are positively related to provincial RRP. Such load increment is significant and even comparable to the daily generation of California, U.S. The provincial daily average electric heating load in SD is estimated to reach 353.23 GWh, accounting for 65.61% of California's daily average generation in 2015.

As illustrated in Fig. 2b, the TCCR of the marginal unit directly influences the carbon emission intensity per electric heating load. As China highly relies on coal for electricity generation, the incremental electric heating loads are generally balanced by marginal coal-fired generators on top of the existing generation resources, except for the cases with renewable energy curtailment. In HN, the generation capacity factor is the lowest among the four provinces, equaling 47.78%. The coal-fired generators have relatively low TCCRs, with a marginal value equaling 275.9 kg/MWh. As a result, the carbon emission intensity per electric heating load in HN is only 397.51 kg/MWh. However, as a large power exporting province, the generation capacity factor in SX is the highest, equaling 66.33%. The marginal TCCR in SX is 368 kg/MWh, and thus the carbon emission intensity per electric heating load can reach as high as 635.71 kg/MWh.

**National impacts of Electric Heating Policy in China**. We extend the base case results in HB, HN, SD, and SX to the other provinces in Northern China, considering 50% of the rural residents substituting electric heating in place of raw coal (Fig. 3a). Our results show three clusters of provincial incremental carbon emissions, which reveals the distribution of rural residents in Northern China (Supplementary Fig. 3).

In the base case, the incremental carbon emission in Northern China in 2015 is estimated to reach 135.60 megatons. Considering the joint uncertainty in PIR and EHM, the emission level may vary from 101.69 to 162.89 megatons (Fig. 3b). The carbon emissions caused by China's EHP are comparable to the annual total emissions in different countries across the world. For example, such incremental carbon emission approximately accounts for 31.02–49.69% of France's annual emission.

Furthermore, the impacts of EHP on China's carbon mitigation in the future are investigated (Fig. 3c). In 2020, we estimate that the incremental carbon emission can reach 168.80 megatons in the base case, and may vary from 130.03 to 197.87 megatons due to PIR and EHM uncertainty. On the other hand, China's urbanization progress will slow down the growth in the carbon emissions caused by EHP. Compared with 2015, the rural population in 2030 is expected to decrease from 48.67 to 32.54% in HB, from 53.15 to 34.00% in HN, from 42.99 to 25.00% in SD, and from 44.98 to 24.59% in SX. While the PIR may further increase to about 90% in 2030, the incremental carbon emission caused by EHP will drop to 119.19–177.47 megatons.

**Techno-economic analysis for low-carbon electric heating pathways**. Renewable energy curtailment has long been a severe issue in China, yielding an enormous waste of clean-energy resources[30]. The national renewable energy curtailment could reach over 80 TWh in 2015, and a lot of wind and solar energy was curtailed in the northern and western provinces in China, including Gansu, Inner Mongolia, Xinjiang, etc. (Supplementary Fig. 4). The interconnected ultrahigh-voltage direct/alternating current (UHVDC/AC) transmission systems provide a natural platform for balancing electric heating load with inter-regional renewable energy[31] (Supplementary Fig. 4). The impacts of matching electric heating load with renewable energy in the four provinces are illustrated in Fig. 4a. With more electric heating loads satisfied by renewable energy, the carbon emissions caused by EHP keep decreasing. To totally offset the incremental carbon emissions, the requirements for annual additional renewable energy in HB, HN, SD, and SX can reach 19.20, 25.21, 36.06, and 12.28 TWh, accounting for 0.60%, 0.71%, 0.70%, and 0.88% of provincial electricity consumption in 2015, respectively. Note that the marginal carbon emission reduction declines due to an increasing curtailment of renewable energy, which is significantly apparent in HB, HN, and SD.

The incremental carbon emissions caused by EHP can be effectively limited by installing distributed photovoltaic (PV) resources (Fig. 4b, Supplementary Fig. 5). In contrast to the case without PV, the incremental carbon emissions in the four provinces after installing 10-kW PV can be reduced by 5.09, 10.35, 14.71, and 8.89 megatons, respectively (Supplementary Fig. 5). However, as illustrated in Fig. 4b, the accommodation capability for PV declines, leading to more solar energy curtailment and thus a decreasing carbon emission reduction rate. Additionally, we discover that provincial carbon emission reduction per capita per kW is highly related to the level of irradiance. The daily average irradiance in SD is 2.88 $kW/m^2$, which is stronger than those in HN and HB. Thus, SD has a higher carbon emission reduction rate. However, in spite of a slightly weaker irradiance in SX, the carbon emission reduction rate in SX is higher than that in SD. This is because SX has a greater carbon emission intensity per electric heating load (Fig. 2b).

As illustrated in Fig. 4c, it is an effective solution to reduce carbon emissions by popularizing HPs instead of EHs. This is because the coefficient of performance (COP) of an HP is generally higher than that of an EH[32]. To totally offset the incremental carbon emission, the requirements for HP proportion are estimated to reach [68.33%, 80.26%] in HB, [63.56%, 74.69%] in HN, [70.67%, 82.86%] in SD, and [78.28%, 91.82%] in SX.

Furthermore, we analyze the average annualized cost per household in 25 cities in Northern China (Fig. 5). Here we

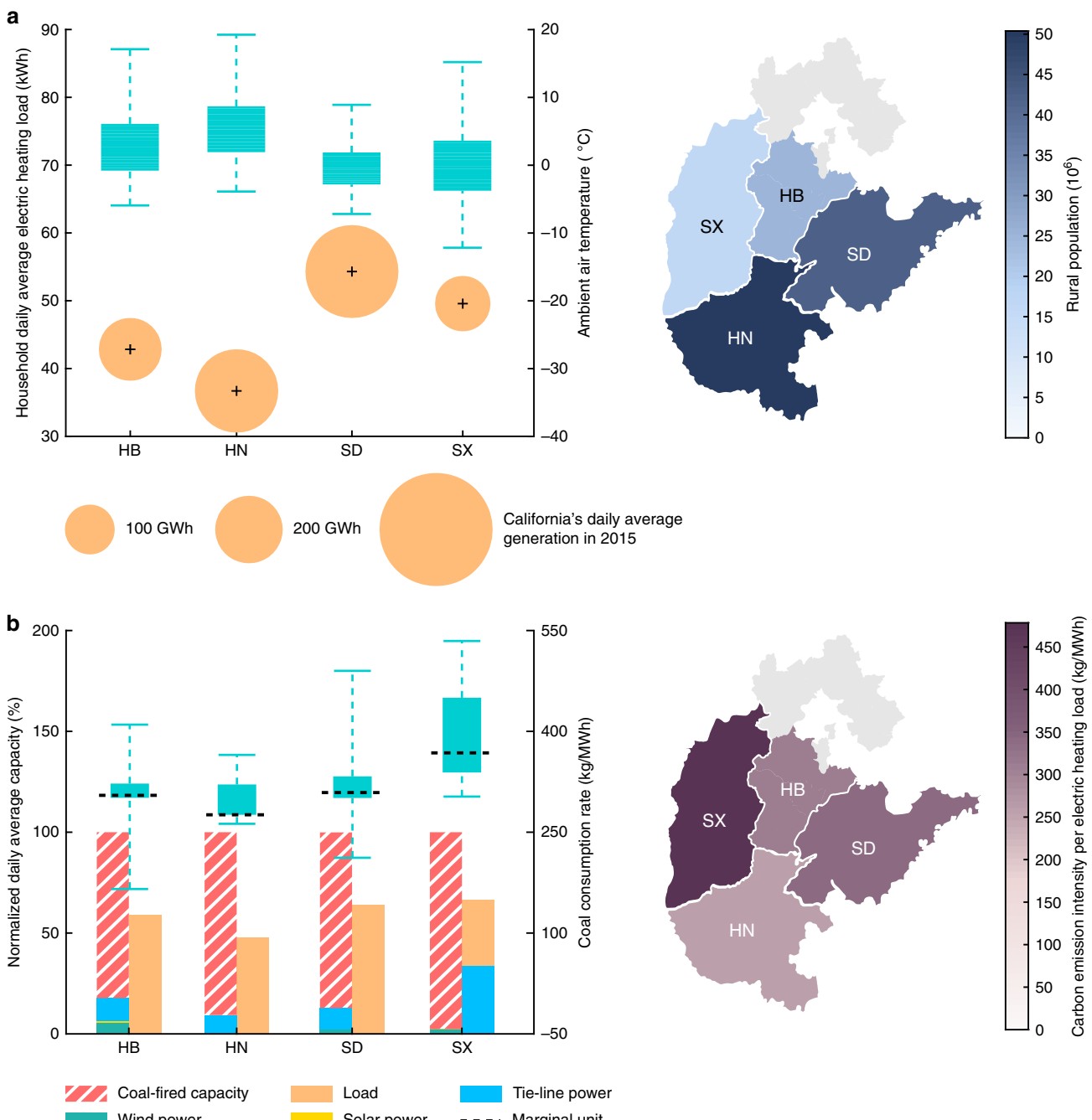

**Fig. 2 Analysis of the key factors influencing provincial carbon emissions caused by EHP. a** Household daily average electric heating load and rural resident population (RRP) in Hebei (HB), Henan (HN), Shandong (SD), and Shanxi (SX) provinces. The left figure shows the relationship between the household daily average electric heating load and hourly ambient air temperature (AAT) in winter. The center of each bubble represents the load of a single household, and the radius represents the provincial daily average electric heating load of all rural residents. Each box shows the distribution of hourly AAT ($n = 1464$). The right figure shows the RRP of four provinces, and the gray area, i.e., the northern region of HB, is excluded from the analysis due to data limitation. **b** Daily average generation capacity factors and carbon emission intensity per electric heating load. In the left figure, the bars represent the normalized generation capacity and electric load before electric heating, and the total available generation capacity is scaled to 100%. Each box shows the distribution of the thermal coal consumption rates (TCCRs) of coal-fired generators ($n = 97$ for HB, $n = 153$ for HN, $n = 190$ for SD and SX). The minimum/ maximum of each box indicates the minimal/maximal value, and the lower and upper percentiles are 25% and 75%, respectively.

compare seven cases, including coal, EHs, HPs, rooftop solar with poverty alleviation program (PAP), high-level subsidy, medium-level subsidy and rooftop solar without subsidy. In China, the solar energy for poverty alleviation program aims to expand over 10 GW distributed PV capacity, benefiting more than 2 million rural households by 2020[33]. The subsidy for PAP is 0.42 ¥/kWh[34].

For other distributed PV systems that are not involved in the PAP, we consider (i) high-level subsidy equaling 0.37 ¥/kWh[34], implemented before 2019; (ii) medium-level subsidy equaling 0.18 ¥/kWh[35], implemented after 2019; and (iii) the case without subsidy. We discover that compared with electric heating, it is still the most cost-saving way for rural residents to burn raw coal for

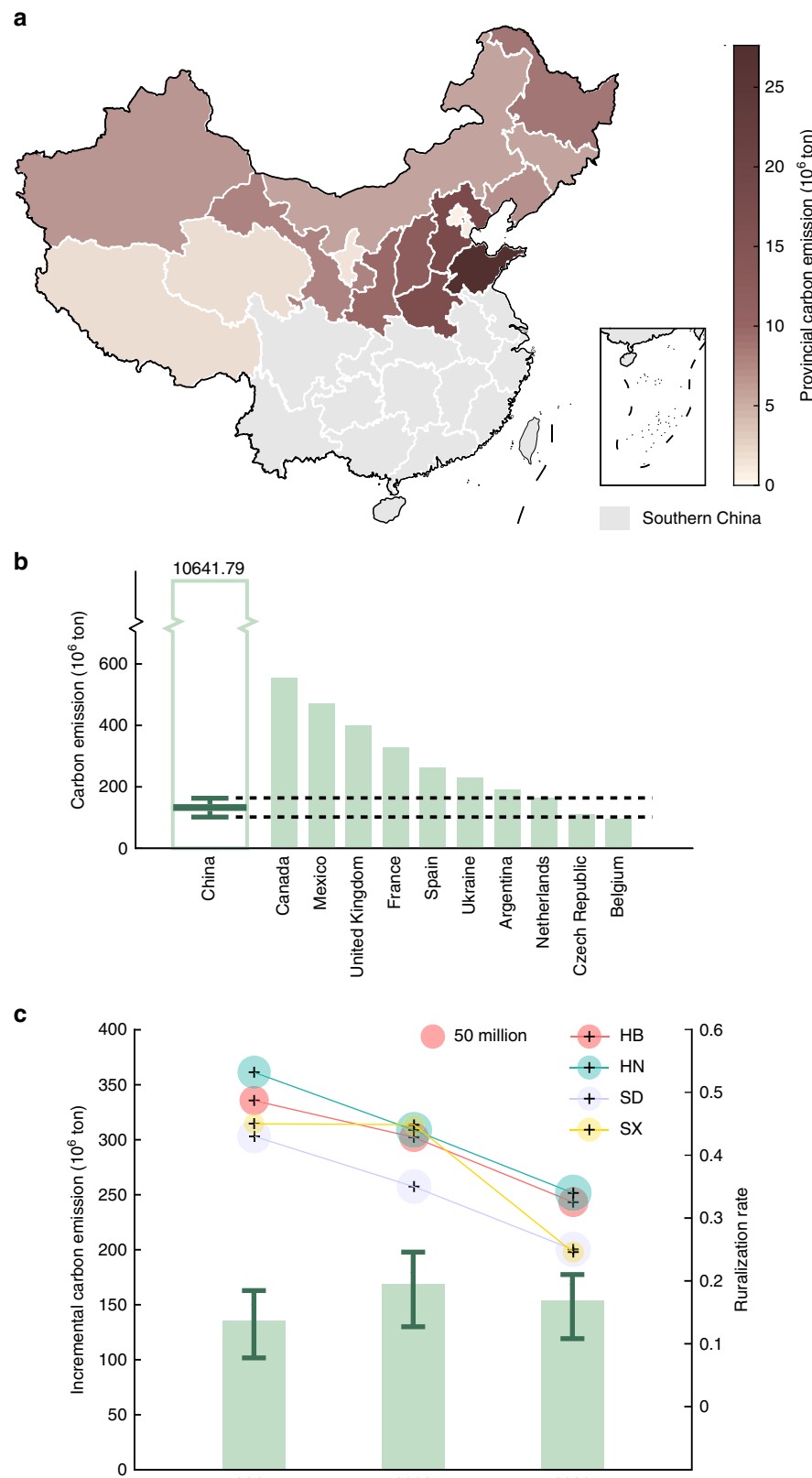

space heating in winter. Due to the relatively low price for raw coal, a household only needs to spend ¥ 633.69–1222.40 in winter to consume 333.52–643.37 kg raw coal.

Additionally, electric heating requires rural residents to pay extra money for investment and electricity bills. In most cities, using electric heaters is a cost-efficient solution because of the low capital costs. The average annualized costs for different households vary from ¥ 1583.30 to ¥ 3500.05, which are much less than those spent by using HPs, i.e., ¥ 3377.21–3990.57. This indicates that the current price of an HP is still too high for a residential household. In contrast to an EH, the cost savings from electricity bills cannot even recover the investment for an HP.

**Fig. 3 National impacts of China's Electric Heating Policy (EHP). a** Provincial carbon emission increase after implementing EHP among 50% of rural residents. The gray areas represent the provinces in Southern China that are excluded from the analysis in this paper. **b** Comparisons of carbon emissions between China and other countries in 2015[56]. The bar represents a country's annual carbon emission in 2015. The box shows the variation of China's incremental carbon emission considering the uncertainty in the policy implementation rate (PIR) and electric heating mix ($n = 36$). The minimum/maximum of the box indicates the minimal/maximal value of national incremental carbon emission, and the lower and upper percentiles are 25% and 75%, respectively. **c** Incremental carbon emissions after implementing EHP in Northern China in 2015, 2020 and 2030. The bars represent the incremental carbon emissions with an average electric heating mix considering the PIR equaling 50% in 2015, 70% in 2020, and 90% in 2030. The error bar defines the range of incremental carbon emission with the PIR varying from 45 to 55% in 2015, from 65 to 75% in 2020, and from 85 to 95% in 2030 ($n = 2$). The radius of each bubble shows the provincial population in Hebei (HB), Henan (HN), Shandong (SD), and Shanxi (SX), and the center indicates the provincial ruralization rate, i.e., the rural population over the total amount.

As illustrated in Fig. 5, the subsidy for solar energy plays an important role in popularizing the distributed PV systems among rural residents. The costs in PV-PAP are the least among solar-powered electric heating cases because of the highest subsidies, and even less than those by using EHs in some cities. In PV-PAP and PV-H where the subsidy is high, the average annualized costs vary within ¥ [1792.77, 3413.05] and ¥ [1975.64, 3605.11], respectively. However, Chinese government announced to reduce the subsidy for solar energy to 0.18 ¥/kWh in 2019. The costs in PV-M and PV-N significantly increase to ¥ [2656.17, 4334.98] and ¥ [3300.87, 5026.42], respectively.

## Discussion

To reduce the pollutant emissions from rural residents, Chinese government has issued Electric Heating Policy to substitute electric heating in place of burning raw coal. However, electric heating can lead to a significant increase in load demands from power grids. We estimate that in the base case with 50% PIR, the load increase in HB, HN, SD, and SX can reach 24.56, 43.27, 53.87, and 19.11 TWh, respectively (Supplementary Fig. 6). The incremental electric loads require more electricity from coal-fired power plants, thus releasing more carbon emissions. Compared with locally burning raw coal, it is less efficient to use EHs for space heating. The generating efficiency of power plants is generally around 40%[36], the power loss on transmission and distribution networks is about 6–10%, and the COP of an EH is ~80%[37]. This indicates that a large fraction of energy is dissipated along electricity generation, transmission, distribution and consumption sectors[38,39]. As a result, substituting 1 kg of raw coal requires 1.89, 1.68, 1.90, and 2.17 kg of thermal coal to satisfy the electric heating load in HB, HN, SD, and SX, respectively (Fig. 2b). On the other hand, the oxidization rate of thermal coal is much higher than that of raw coal, indicating that thermal coal has a higher carbon emission factor[40]. Therefore, in spite of an effective raw coal reduction among rural residents, China's EHP can lead to significant carbon emissions released from the power sector.

It should be noted that both power transmission and distribution networks are not incorporated in our theoretical model due to data limitation, which may underestimate future curtailment of renewable energy. Therefore, we claim that this paper provides a conservative estimation for incremental carbon emissions induced by China's EHP. The impact of network congestion on renewable energy curtailment and the associated carbon emissions deserves an in-depth investigation in future work.

Two low-carbon electric heating pathways are suggested for China and other countries with similar situations[41,42], i.e., balancing electric heating load with renewable energy, and improving the efficiency of electric heating. Specifically, the carbon emission increase caused by EHP can be effectively offset by integrating the interprovincial renewable energy. In 2018, for example, 85.42 GWh of electric heating load was directly satisfied by wind and solar stations in SX[43]. However, the marginal carbon

emission reduction gets low with the increase in renewable energy penetration, which is validated in both cases with province-level renewable energy and distributed PV resources (Fig. 4a, b). On the other hand, to totally offset the carbon emissions induced by EHP, the proportion of HPs is estimated to increase to ~60–90% in Northern China (Fig. 4c).

In the past several years, Chinese government has encouraged rural residents in Northern China to switch to electric heating by offering subsidies. In HB, for example, a household can be subsidized with ¥ 7400 (about $ 1000) in a one-off scheme to invest in electric heating devices[44]. Additionally, the subsidy for electric heating load can reach 0.12 ¥/kWh[44], approximately accounting for 20–25% of the retail tariff. In spite of such profitable policy, electric heating is still too expensive for rural residents in Northern China. The average annualized costs per household for using EHs and HPs are estimated to reach ¥ [1583.30, 3500.05] and ¥ [3377.21, 3990.57], respectively, much more than those for burning raw coal, i.e., ¥ [633.69, 1222.40]. In addition, we claim that financial subsidy can yield a great impact on the annualized costs for using PVEH. The costs in PV-PAP can even reach less than those by using EHs in some cities, e.g., Chengshan, Longkou, Huixian (Fig. 5). In SD, the costs in PV-PAP are less than those by using EHs because the AATs in these cities are relatively low and high electric heating load is required, leading to extra electricity bills. However, in the southern cities in HN where the AATs are generally high, EHs show a cost-efficient advantage over PV-PAP.

In this paper, we summarize three policy suggestions for China and other developing countries. First, the government must explore the potential incompatibility between any new policy and the existing ones. According to our analyses, the underestimation of the greenhouse effect caused by EHP can impede China's carbon mitigation process in the future. Meanwhile, an increasing penetration of electric heating may lead to the shortage of generation capacity and flexible load-following resources, thus threatening the secure and reliable operation of power grids. Second, the government is suggested to match large-scale renewable generation with electric heating load. Considering the relatively high capital costs for HPs and PV, it is better to accommodate the surplus renewable energy in China's northern and western provinces. Note that the Shanxi government has gained success in organizing the bilateral trading between wind/solar stations and some villages using EHs for heating. However, we also suggest that the carbon reduction performance of developing interprovincial renewable energy and distributed solar systems should be systematically evaluated in the future considering potential network congestion. Third, we suggest that the government should provide adequate incentives to encourage electric heating among rural residents. In China, the current capital cost for an HP is still too high for rural residential space heating. The one-off subsidy and electricity bill discount are insufficient to popularize HPs as dominant heating devices.

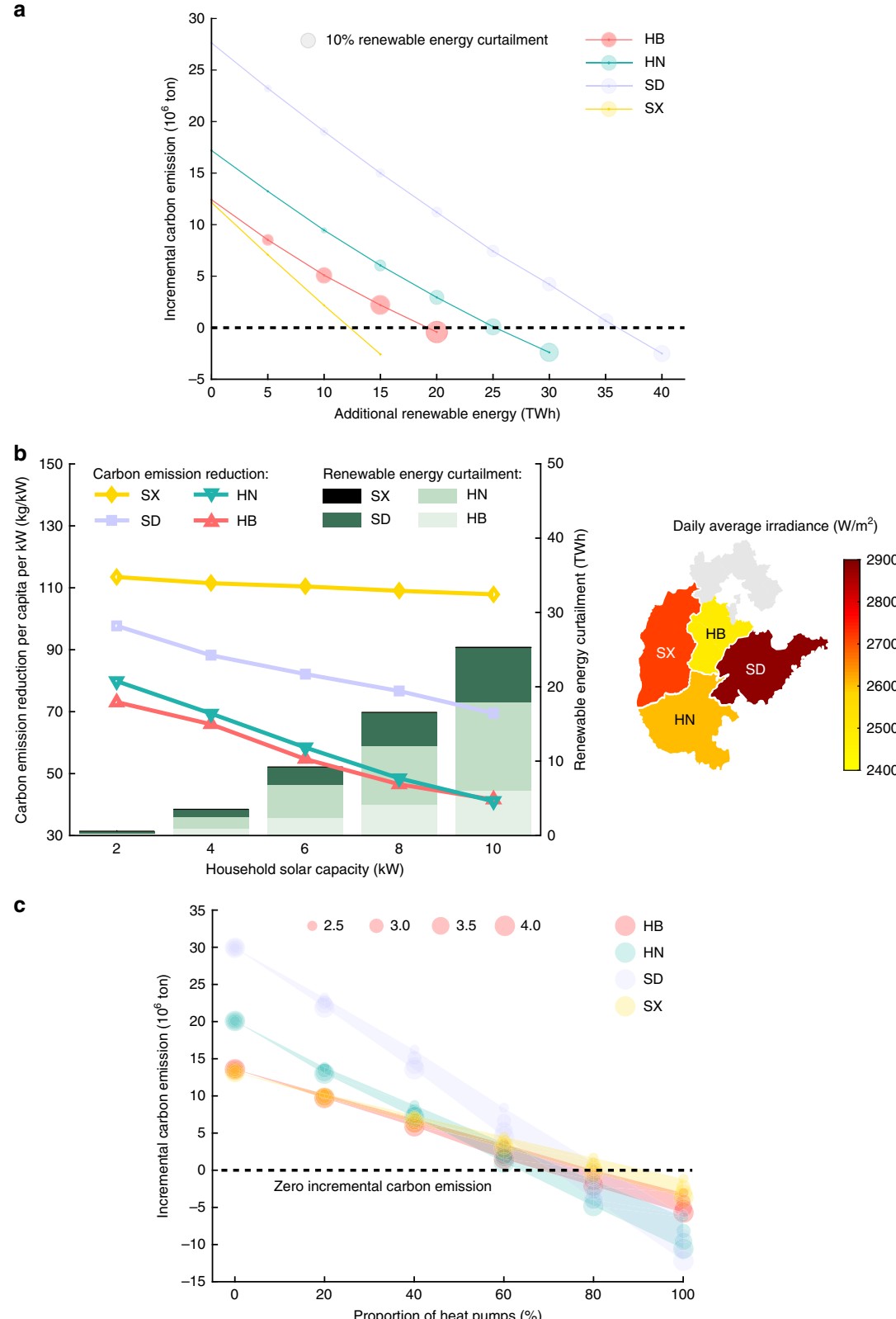

**Fig. 4 Impacts of the integration of renewable energy and the improvement of electric heating devices. a** Carbon emissions in Hebei (HB), Henan (HN), Shandong (SD), and Shanxi (SX) with the integration of renewable energy. The radius of each bubble represents the curtailment rate for additional renewable energy, and the center indicates the incremental carbon emission caused by electric heating. **b** Carbon emission reduction capability of distributed PV in the four provinces. The lines represent the carbon emission reduction per capita per kW, and the bars show the curtailment of solar energy. **c** Relationship between carbon emission and the proportion of heat pumps (HPs) used for electric heating. The radius of each bubble represents the coefficient of performance (COP) of an HP, and the shadows are bounded by the lines with COP equaling 2.5 and 4.0.

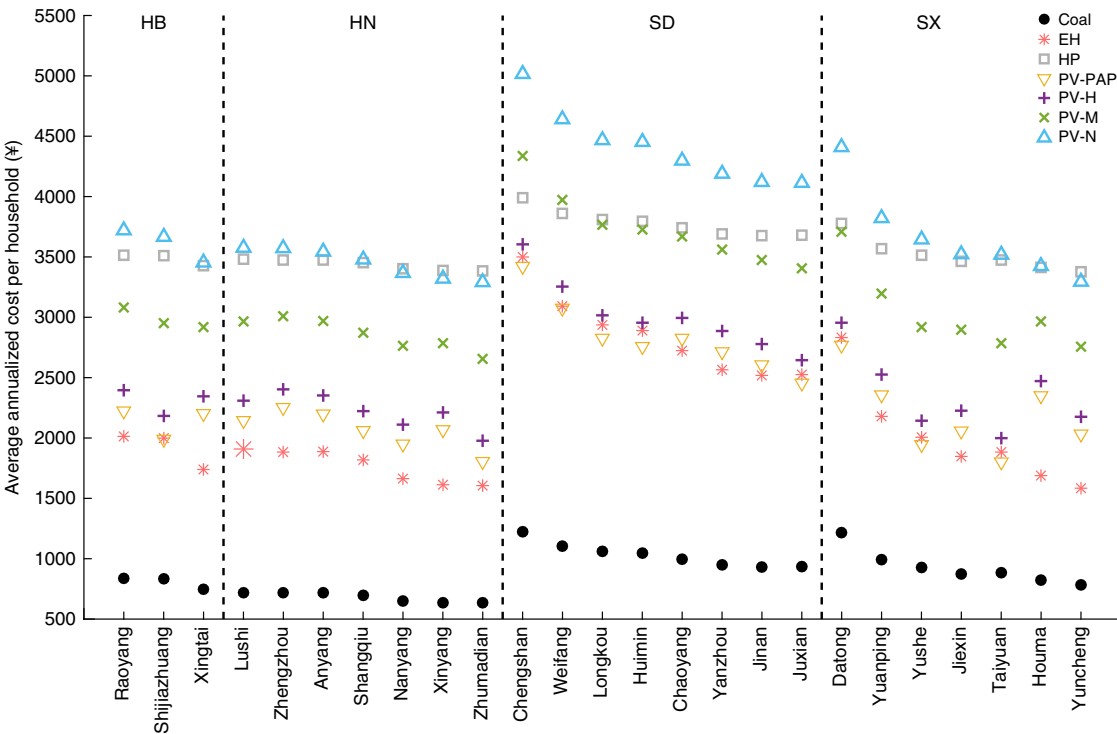

**Fig. 5 Average annualized cost per household in different cities in Northern China.** "Coal" represents that a rural household burns raw coal for space heating in winter. "EH" and "HP" represent that a rural household uses an 8-kW electric heater and heat pump, respectively. "PV-PAP", "PV-H", "PV-M", and "PV-N" represent that a rural household uses solar-powered electric heating with a 3-kW solar panel and 8-kW electric heater, and the subsidies for solar energy are 0.42, 0.37, 0.18, and 0 ¥/kWh, respectively. Note that "PAP", "H", "M", and "N" are short for poverty alleviation program, high-level subsidy, medium-level subsidy and no subsidy.

## Methods

**Rural resident data.** In this paper, we simulate the carbon emissions in four provinces in Northern China, i.e., Hebei, Henan, Shandong and Shanxi. To assess the emissions caused by rural space heating, the following data are needed: rural resident population, housing areas, household heating coal consumption, household electric heating load, and rooftop solar power.

According to the Sixth National Census in China, the 2015 rural populations in HB, HN, SD, and SX were 36.14, 50.39, 42.33, and 16.48 million, respectively. The 2015 rural population in 16 provinces (autonomous regions and municipalities) in Northern China was 256.19 million. According to the population target planning in 2020, the rural populations in the HB, HN, SD, and SX will be 33.00, 49.06, 35.88, and 16.60 million, respectively. In 2030, the rural populations in the four provinces are expected to drop to 25.74, 39.10, 26.67, and 9.67 million, respectively.

The housing area of a family is a key factor that determines the heating coal consumption and electric heating load. The housing area data are collected from "Report on Chinese Residential Energy Consumption", published by the National Academy of Development and Strategy, Renmin University of China[45]. The housing areas are divided into eight intervals, i.e., [15,30], (30,50], (50,70], (70,90], (90,120], (120,150], (150,180], and (180,250] m², accounting for 1.05%, 3.48%, 7.67%, 13.24%, 24.39%, 16.72%, 14.29%, and 19.16%, respectively.

Given outdoor air temperature, indoor comfort temperature and the housing area as input, a household's heating coal consumption and electric heating load can be simulated by using EnergyPlus, a building energy consumption simulation software developed by Lawrence Berkeley National Laboratory (LBNL) and some other institutions, sponsored by the Department of Energy[46]. The outdoor air temperature data are collected from the weather dataset arranged by the World Meteorological Organization[47]. In HB, we use the weather data from three cities, i.e., Raoyang, Shijiazhuang and Xingtai. In HN, we use the weather data from seven cities, including Anyang, Lushi, Nanyang, Shangqiu, Xinyang, Zhengzhou, and Zhumadian. In SD, we use the weather data from eight cities, including Chaoyang, Chengshantou, Huimin, Jinan, Juxian, Longkou, Weifang, and Yanzhou. In SX, we use the weather data from seven cities, including Datong, Houma, Jiexiu, Taiyuan, Yuanping, Yuncheng, and Yushe. According to the "Indoor Air Quality Standard" recognized by Chinese government, the indoor air temperature of a rural household is required to reach at least 13–17 °C, which is set as the input of indoor comfort temperature.

In this paper, we collect hourly residential solar power data in the aforementioned cities in the four provinces from the National Renewable Energy

Laboratory (NREL)[48]. The average daily generation of a 1-kW solar panel in HB, HN, SD, and SX is 2.07, 2.14, 2.47, and 2.28 kWh, respectively.

**Electric power system data.** To assess the thermal coal consumption and associated carbon emissions of China's electric power systems, the following data are needed: thermal generator parameters, renewable power, and electric power system load.

We collect the parameters of thermal generators in HB, HN, SD, and SX provinces in 2015. In the southern region of HB power grid, there are 97 thermal generators, with the median TCCR equal to 306.11 kg/MWh. The total installed capacity of thermal generators is 28.66 GW, and the median value is 330 MW. In HN power grid, there are 153 thermal generators, with the median TCCR equal to 297.80 kg/MWh. The total installed capacity is 61.49 GW, and the median value is 320 MW. In SD power grid, there are 190 thermal generators, with the median TCCR equal to 314.15 kg/MWh. The total installed capacity is 62.41 GW, and the median value is 320 MW. In SX power grid, there are 190 thermal generators, with the median TCCR equal to 368.00 kg/MWh. The total installed capacity is 56.46 GW, and the median value is 300 MW. Additionally, the installed capacities of wind farms in HB, HN, SD, and SX are 10.22, 0.91, 7.21, and 6.99 GW, respectively. The installed capacities of solar stations in HB, HN, SD, and SX are 2.22, 0.41, 1.33, and 1.11 GW, respectively.

Due to data limitation, we collect the provincial electric loads measured in hours from 11/01/2015 to 12/31/2015 in HB, HN, SD, and SX. The total electric loads during the two months in the four provinces are 30.15, 47.44, 67.02, and 27.58 TWh, respectively. In this paper, we estimate the carbon emissions during a heating season to be 2.5 times those during the two months because a heating season generally lasts from November 1 to March 31 in the next year.

In 2015, the national installed capacities of thermal, wind and solar generation are 1.01, 0.13, and 0.04 TW, respectively. According to "China Energy & Electricity Outlook" published by State Grid Energy Research Institute (SGERI), such capacities will be 1.19, 0.28, and 0.28 TW in 2020, respectively[49]. In 2030, the capacities are expected to reach 1.53, 0.70, and 0.56 TW, respectively[49]. In addition, the national electric load demands in 2015, 2020, and 2030 are $5.7 \times 10^3$, $7.7 \times 10^3$, and $11.1 \times 10^3$ TWh, respectively[49].

**Rural heating assessment.** In this paper, household heating coal consumption and electric heating load are simulated by EnergyPlus. When simulating a

household's heating coal consumption, we set the heating coil type as gas and transform the gas consumption into coal consumption based on the total amount of heat. The transformation is expressed as follows:

$$Q_i^{\text{HCoal}} = H_i^{\text{Gas}}/h^{\text{Coal}}, \tag{1}$$

where $Q_i^{\text{HCoal}}$ is the heating coal consumption of the $i$th household; $H_i^{\text{Gas}}$ represents the total amount of heat produced by burning gas; and $h^{\text{Coal}}$ is the heating value of coal, i.e., $2.93 \times 10^7$ J/kg. When we set the heating coil type as electricity, a household's hourly electric heating load can be directly acquired by running EnergyPlus. For electric heaters, we set the efficiency of the electric heating coil as 80%. For air-sourced heat pumps, we change the efficiency from 250 to 400%[50].

To systematically evaluate the households' heating energy consumption considering different sizes, we conduct sensitivity analyses on the sizes of houses. We scan the length, width and height of houses from 5 to 20 m, from 3 to 12 m, and from 3 to 5 m, respectively. Then, we categorize the houses with different sizes into eight intervals based on housing areas, i.e., [15,30], [30,50], [50,70], [70,90], [90,120], [120,150], [150,180], and [180,250] m². Let $\bar{Q}_j^{\text{HCoal}}$ and $\bar{P}_j^{\text{Elec}}(t)$ be the average heating coal consumption and hourly electric heating load at time slot $t$ of the households in the $j$th area interval, respectively (Supplementary Fig. 7). The average household heating coal consumption $\bar{Q}^{\text{HCoal}}$ and the average household hourly electric heating load $\bar{P}^{\text{Elec}}(t)$ in a province can be calculated as follows:

$$\bar{Q}^{\text{HCoal}} = \sum_{j=1}^{8} \gamma_j \bar{Q}_j^{\text{HCoal}}, \tag{2}$$

$$\bar{P}^{\text{Elec}}(t) = \sum_{j=1}^{8} \gamma_j \bar{P}_j^{\text{Elec}}(t), \tag{3}$$

where $\gamma_j, j = 1, 2...8$ represents the proportion of the $j$th area interval, i.e., 1.05%, 3.48%, 7.67%, 13.24%, 24.39%, 16.72%, 14.29%, and 19.16%, respectively. Then a province's total heating coal consumption $Q^{\text{HCoal}}$ and total hourly electric heating load $P^{\text{Elec}}(t)$ can be obtained by using the following equations:

$$Q^{\text{HCoal}} = \bar{Q}^{\text{HCoal}} \times \rho \times p^{\text{PIR}}, \tag{4}$$

$$P^{\text{Elec}}(t) = \bar{P}^{\text{Elec}}(t) \times \rho \times p^{\text{PIR}}, \tag{5}$$

where $\rho$ is provincial rural resident population; and $p^{\text{PIR}}$ represents the policy implementation rate.

In addition, a household's net electric heating load $P_i^{\text{Net}}(t)$ is calculated as the difference between the load $P_i^{\text{Elec}}(t)$ and rooftop solar power $P_i^{\text{Solar}}(t)$:

$$P_i^{\text{Net}}(t) = P_i^{\text{Elec}}(t) - P_i^{\text{Solar}}(t). \tag{6}$$

**Electricity dispatch model.** In this paper, a day-ahead unit commitment model is formulated to quantify the thermal coal consumption. The mathematical formulation is shown as follows:

$$\min_{\mathbf{X}} \sum_{i \in \Phi^{\text{CG}}} \sum_{t \in \Phi^{\text{T}}} [c_i^{\text{CG}} P_i^{\text{CG}}(t) + c_i^{\text{U}} Y_i(t) + c_i^{\text{D}} Z_i(t)], \tag{7}$$

subject to

$$\sum_{i \in \Phi^{\text{CG}}} P_i^{\text{CG}}(t) + \sum_{j \in \Phi^{\text{RG}}} P_j^{\text{RG}}(t) + \sum_{k \in \Phi^{\text{TL}}} P_k^{\text{TL}}(t) = P^{\text{Load}}(t) = P^{\text{Orig}}(t) + P^{\text{Net}}(t), \forall t \in \Phi^{\text{T}}, \tag{8}$$

$$\sum_{i \in \Phi^{\text{CG}}} P_{i,\max}^{\text{CG}} U_i(t) \geq R(t), \forall t \in \Phi^{\text{T}}, \tag{9}$$

$$P_{i,\min}^{\text{CG}} U_i(t) \leq P_i^{\text{CG}}(t) \leq P_{i,\max}^{\text{CG}} U_i(t), \forall i \in \Phi^{\text{CG}}, \forall t \in \Phi^{\text{T}}, \tag{10}$$

$$Y_i(t) + Z_i(t) \leq 1, \forall i \in \Phi^{\text{CG}}, \forall t \in \Phi^{\text{T}}, \tag{11}$$

$$Y_i(t) - Z_i(t) = U_i(t) - U_i(t-1), \forall i \in \Phi^{\text{CG}}, \forall t \in \Phi^{\text{T}}, \tag{12}$$

$$\sum_{\delta=t}^{t+T_i^{\text{U}}-1} U_i(\delta) \geq T_i^{\text{U}} Y_i(t), \forall i \in \Phi^{\text{CG}}, \forall t \in \Phi^{\text{T}}, \tag{13}$$

$$\sum_{\delta=t}^{t+T_i^{\text{D}}-1} [1 - U_i(\delta)] \geq T_i^{\text{D}} Z_i(t), \forall i \in \Phi^{\text{CG}}, \forall t \in \Phi^{\text{T}}, \tag{14}$$

$$P_i^{\text{CG}}(t) \in \mathbb{R}^+ \cup \{0\}, \forall i \in \Phi^{\text{CG}}, \forall t \in \Phi^{\text{T}}, \tag{15}$$

$$U_i(t), Y_i(t), Z_i(t) \in \{0, 1\}, \forall i \in \Phi^{\text{CG}}, \forall t \in \Phi^{\text{T}}, \tag{16}$$

where the decision variables are denoted by $\mathbf{X}$, including the hourly power of coal-fired generators $P_i^{\text{CG}}(t)$, the on/off states of coal-fired generators $U_i(t)$, and the

startup/shutdown variables of coal-fired generators $Y_i(t)/Z_i(t)$. As shown in Eq. (7), the proposed unit commitment model is aimed at minimizing the generation costs $c_i^{\text{CG}} P_i^{\text{CG}}(t)$, the startup and shutdown costs, $c_i^{\text{U}} Y_i(t)$ and $c_i^{\text{D}} Z_i(t)$, of all coal-fired generators over a 24-h time horizon. $\Phi^{\text{CG}}$ is the set of coal-fired generators, and $\Phi^{\text{T}}$ is the set of hours. $c_i^{\text{CG}}$ is the cost per MWh of the $i$th coal-fired generator, and $c_i^{\text{U}}$ and $c_i^{\text{D}}$ are the startup and shutdown costs per time of the $i$th coal-fired generator. Equation (8) is the balance for power supply and load demand, where $P_j^{\text{RG}}(t)$ and $P_k^{\text{TL}}(t)$ are the power of the $j$th renewable generator and the $k$th interchange tie-line; the system total load $P^{\text{Load}}(t)$ consists of two parts, $P^{\text{Orig}}(t)$ and $P^{\text{Net}}(t)$, i.e., the original system load and the total net electric heating load in Eq. (6); $\Phi^{\text{RG}}$ and $\Phi^{\text{TL}}$ are the sets of renewable generators and interchange tie-lines, respectively. Constraint (9) shows the spinning reserve requirement, where $P_{i,\max}^{\text{CG}}$ is the installed capacity of the $i$th coal-fired generator, and $R(t)$ is the reserve requirement at time slot $t$. Constraint (10) shows the lower and upper limits for coal-fired generators' power, where $P_{i,\min}^{\text{CG}}$ is the minimal power when the $i$th coal-fired generator is online. Constraints (11) and (12) show the relationships between $U_i(t)$, $Y_i(t)$, and $Z_i(t)$. Constraints (13) and (14) are the minimum on/off hours of coal-fired generators, where $T_i^{\text{U}}$ and $T_i^{\text{D}}$ are the minimum on and off hours of the $i$th coal-fired generator. In (15) and (16), the bounds of the decision variables are defined.

By optimizing the unit commitment model in a day-ahead rolling manner, the daily optimal scheduling strategies for coal-fired generators $\mathbf{P}^{\text{CG*}}$, $\mathbf{U}^*$, $\mathbf{Y}^*$, and $\mathbf{Z}^*$ can be obtained, which set up the operation plans of generators. Let $C_d^{\text{UC}}$ be the optimal objective value of the unit commitment model for the $d$th day, representing the daily costs of thermal coal. Thus, the total thermal coal consumption during $N$ days can be calculated as follows:

$$Q^{\text{GCoal}} = \sum_{d=1}^{N} (C_d^{\text{UC}}/\lambda^{\text{Coal}}), \tag{17}$$

where $Q^{\text{GCoal}}$ is the total thermal coal consumption, and $\lambda^{\text{Coal}}$ is the price of thermal coal.

Note that in the power balance constraint (8), the renewable power, the interchange tie-line power and the system load are collected from power grid companies in four provinces. Given a renewable power and electric heating scenario, we can input the renewable power data and net electric heating load data into the unit commitment model accordingly, and obtain the associated thermal coal consumption. For example, for the scenario with additional renewable energy, we proportionally expand the hourly renewable power data. Based on these data, the unit commitment model is optimized and the total thermal coal consumption for this scenario can be obtained. Due to data limitation, power transmission and distribution networks are not incorporated in the electricity dispatch model.

**National carbon emission estimation.** In this paper, we simulate the heating coal consumption $Q^{\text{HCoal}}$ and electric heating load $P^{\text{Elec}}$ by EnergyPlus in HB, HN, SD, and SX provinces. Due to data limitation, we estimate the total coal consumption in Northern China by expanding the results of the four provinces.

To estimate the heating coal consumption in Northern China, we firstly calculate the per capita heating coal consumption in the four provinces, which is assumed to equal that in Northern China:

$$\bar{Q}_{\text{NC}}^{\text{HCoal}} = \frac{Q_{\text{HB}}^{\text{HCoal}} + Q_{\text{HN}}^{\text{HCoal}} + Q_{\text{SD}}^{\text{HCoal}} + Q_{\text{SX}}^{\text{HCoal}}}{(1 - p^{\text{PIR}})(\rho_{\text{HB}} + \rho_{\text{HN}} + \rho_{\text{SD}} + \rho_{\text{SX}})}, \tag{18}$$

where $\bar{Q}_{\text{NC}}^{\text{HCoal}}$ is the per capita heating coal consumption in Northern China. $Q_{\text{HB}}^{\text{HCoal}}$, $Q_{\text{HN}}^{\text{HCoal}}$, $Q_{\text{SD}}^{\text{HCoal}}$, and $Q_{\text{SX}}^{\text{HCoal}}$ are the provincial heating coal consumption in HB, HN, SD, and SX, respectively. $\rho_{\text{HB}}$, $\rho_{\text{HN}}$, $\rho_{\text{SD}}$, and $\rho_{\text{SX}}$ represent the rural population in HB, HN, SD, and SX, respectively. Then the total heating coal consumption in Northern China is estimated as follows:

$$Q_{\text{NC}}^{\text{HCoal}} = \bar{Q}_{\text{NC}}^{\text{HCoal}} \times \rho_{\text{NC}} \times (1 - p^{\text{PIR}}), \tag{19}$$

where $Q_{\text{NC}}^{\text{HCoal}}$ is the heating coal consumption in Northern China. $\rho_{\text{NC}}$ represents the rural population in Northern China.

To estimate the thermal coal consumption in Northern China, we firstly calculate the per capita electric heating load in the four provinces, which is assumed to equal that in Northern China:

$$\bar{P}_{\text{NC}}^{\text{Elec}} = \frac{P_{\text{HB}}^{\text{Elec}} + P_{\text{HN}}^{\text{Elec}} + P_{\text{SD}}^{\text{Elec}} + P_{\text{SX}}^{\text{Elec}}}{p^{\text{PIR}}(\rho_{\text{HB}} + \rho_{\text{HN}} + \rho_{\text{SD}} + \rho_{\text{SX}})}, \tag{20}$$

where $\bar{P}_{\text{NC}}^{\text{Elec}}$ is the per capita electric heating load in Northern China. $P_{\text{HB}}^{\text{Elec}}$, $P_{\text{HN}}^{\text{Elec}}$, $P_{\text{SD}}^{\text{Elec}}$, and $P_{\text{SX}}^{\text{Elec}}$ are the provincial electric heating load in HB, HN, SD, and SX, respectively. Then the total electric heating load in another province in Northern China is estimated as follows:

$$P_x^{\text{Elec}} = \bar{P}_{\text{NC}}^{\text{Elec}} \times \rho_x \times p^{\text{PIR}}, \tag{21}$$

where subscript $x$ represents a province in Northern China. In 2015, the rural populations in Heilongjiang, Jilin, Liaoning, Beijing, Tianjin, Inner Mongolia, Shaanxi, Ningxia, Gansu, Xinjiang, and Tibet are 15.70, 12.30, 14.31, 2.93, 2.69, 9.97, 17.48, 2.99, 14.77, 2.92, 12.54, and 2.34 million, respectively. The thermal coal

consumption caused by electric heating load in province $x$ is calculated as follows:

$$Q_x^{\mathrm{GCoal}} = P_x^{\mathrm{Elec}} \times a_x, \qquad (22)$$

where $a_x$ represents the provincial average thermal coal consumption rate, measured in kg/MWh (Supplementary Fig. 8). Therefore, the total thermal coal consumption caused by electric heating load in Northern China $Q_{\mathrm{NC}}^{\mathrm{GCoal}}$ can be estimated as follows:

$$Q_{\mathrm{NC}}^{\mathrm{GCoal}} = \sum_{x \in \Phi^{\mathrm{NC}}} Q_x^{\mathrm{GCoal}}, \qquad (23)$$

where $\Phi^{\mathrm{NC}}$ represents the set of the provinces in Northern China. Given Eqs. (18)–(23), we can estimate the total amount of heating coal and thermal coal in Northern China.

The heating value of standard coal is $h^{\mathrm{Coal}} = 2.93 \times 10^7$ J/kg, and the net carbon content per energy is $\alpha^{\mathrm{Coal}} = 26.59$ tC/TJ[40]. In this paper, the oxidization rate of raw coal is set as $o^{\mathrm{HCoal}} = 83.7\%$, and that of thermal coal is set as $o^{\mathrm{GCoal}} = 99\%$[40]. Therefore, the emission factors of raw coal and thermal coal, denoted by $e^{\mathrm{HCoal}}$ and $e^{\mathrm{GCoal}}$, are calculated below:

$$e^{\mathrm{HCoal}} = o^{\mathrm{HCoal}} \times h^{\mathrm{Coal}} \times \alpha^{\mathrm{Coal}} = 2.39 \, \mathrm{tCO_2/t}, \qquad (24)$$

$$e^{\mathrm{GCoal}} = o^{\mathrm{GCoal}} \times h^{\mathrm{Coal}} \times \alpha^{\mathrm{Coal}} = 2.83 \, \mathrm{tCO_2/t}. \qquad (25)$$

The carbon emissions from the rural residential heating coal and thermal coal consumption in Northern China are shown as follows:

$$E_{\mathrm{NC}}^{\mathrm{C}} = e^{\mathrm{HCoal}} Q_{\mathrm{NC}}^{\mathrm{HCoal}} + e^{\mathrm{GCoal}} Q_{\mathrm{NC}}^{\mathrm{GCoal}}, \qquad (26)$$

where $E_{\mathrm{NC}}^{\mathrm{C}}$ is the $CO_2$ emitted from power generation and rural raw coal for space heating in Northern China.

**Cost analysis for various electric heating pathways**. In this paper, we analyze the average annualized cost per household in 25 cities in HB, HN, SD, and SX provinces. The price of raw coal is estimated to be 0.76 ¥/kg in 2015[51]. A household's annualized cost consists of two fractions, namely the investment cost and the electricity bill.

The annualized investment cost is calculated as follows:

$$\bar{C}^{\mathrm{I}} = \left( C^{\mathrm{I}} - S^{\mathrm{I}} \right) r / [1 - (1 + r)^{-PP}], \qquad (27)$$

where $\bar{C}^{\mathrm{I}}$ and $C^{\mathrm{I}}$ are the annualized and one-off investment costs, respectively. $S^{\mathrm{I}}$ is the one-off subsidy for electric heating, equaling ¥ 7400 (about $ 1000) in Northern China[44]. $r$ is the interest rate, equaling 4.85%[52], and $PP$ is the payback period, which is assumed to be 10 years.

Note that the one-off investment cost $C^{\mathrm{I}}$ varies with different electric heating devices. In Northern China, we estimate that the average electric heating capacity is 8 kW. The one-off investment for an 8-kW EH is ¥ 2000, and that for an 8-kW HP is ¥ 28,000[53]. Additionally, the average rooftop PV capacity of rural residents in China is set to 3 kW[54], and the one-off investment cost for PV is 5000 ¥/kW[55].

For the households with EHs or HPs, the electricity bill during the heating season is calculated as follows:

$$\bar{C}^{\mathrm{E}} = \sum_{t \in \Phi^{\mathrm{HS}}} P^{\mathrm{Elec}}(t) \times (\pi^{\mathrm{R}} - S^{\mathrm{E}}), \qquad (28)$$

where $\bar{C}^{\mathrm{E}}$ is the electricity bill caused by electric heating. $\pi^{\mathrm{R}}$ is the retail tariff, equaling 0.36, 0.38, 0.39, and 0.33 ¥/kWh in HB, HN, SD, and SX, respectively. $S^{\mathrm{E}}$ is the discount tariff for electric heating, equal to 0.12 ¥/kWh[44]. Note that each household is settled on an hourly basis in Eq. (28), where $\Phi^{\mathrm{HS}}$ is the set of the hours during the heating season and $P^{\mathrm{Elec}}(t)$ is the hourly electric heating load.

For the households with PVEH, the electricity bill is calculated as follows:

$$\bar{C}^{\mathrm{E}} = \sum_{t \in \Phi^{\mathrm{HS}}} P_+^{\mathrm{Elec}}(t) \times (\pi^{\mathrm{R}} - S^{\mathrm{E}}) - \sum_{t \in \Phi^{\mathrm{Y}}} P^{\mathrm{Solar}}(t) \times S^{\mathrm{PV}}, \qquad (29)$$

where $P_+^{\mathrm{Net}}(t)$ is the net electric heating load after taking the positive value, i.e., $P_+^{\mathrm{Net}}(t) = \max\{0, P^{\mathrm{Elec}}(t) - P^{\mathrm{Solar}}(t)\}$. $\Phi^{\mathrm{Y}}$ is the set of the hours in a year, i.e., 1, 2,…, 8760. $S^{\mathrm{PV}}$ represents the subsidy for hourly solar generation, e.g., $S^{\mathrm{PV}} = 0.42$ ¥/kWh for PVP.

**Reporting summary**. Further information on research design is available in the Nature Research Reporting Summary linked to this article.

## Data availability

Outdoor air temperature data are available from the dataset arranged by the World Meteorological Organization ([https://www.energyplus.net/weather]). Solar irradiance and power generation data are available from the National Renewable Energy Laboratory ([https://pvwatts.nrel.gov]). Source data are provided with this paper.

## Code availability

The code used in this study is available from the authors upon request.

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

## Acknowledgements

This work was supported in part by the Major Smart Grid Joint Project of National Natural Science Foundation of China and State Grid (No. U1766212), in part by the State Grid Corporation of China (1100-201957275A-0-0-00), and in part by Institute for National Governance and Global Governance, Tsinghua University.

## Author contributions

J.W., H.Z., Q.X., and C.K. conceived and designed the research. J.W., H.Z., and Z.Y. developed the framework and formulated the theoretical model. H.Z. and D.M.K. wrote the introduction. J.W., H.Z., Z.M., and Z.L. carried out the data search. J.W. and M.W. carried out the simulations and analyses. J.W., D.M.K., and Z.M. conducted the analysis of the integrated strategy. All authors contributed to the discussions on the method and the writing of this article.

## Competing interests

The authors declare no competing interests.
