## [Peer Review File · Nature Communications]

Reviewers' comments:

Reviewer #1 (Remarks to the Author):

Wang et al. performed an analysis of the impacts of China's Electric Heating Policy (EHP) on the carbon dioxide and air pollutant emissions. This is a typically multi-disciplinary study relating energy (e.g., electricity, coal and renewable energy), environment (e.g., carbon and air pollutant) and economy (e.g., cost). It looks that the authors are familiar with the energy system. But there are some problems when they extend the results to the field of environment. For example, the authors use the same CO₂ emission factor of coal used in power plant and residential heating. But, the coal combustion of power plant is generally more sufficient (i.e., the oxidation rate is higher), so the carbon emission level of 1kg coal in power plant is higher than that of residential heating. Another example is NO₂, which comes from the reaction of N₂ and O₂ in the atmosphere at high temperature (content of N in fossil fuel is very low). The authors didn't seem to realize these problems.

The authors conclude that the EHP will significantly increase the CO₂ emissions but reduce air pollutant emissions (e.g., SO₂). I am not surprised about this conclusion because the EHP policy was designed to mitigate the air pollution in China. Especially in China's rural area, the EHP could significantly reduce the serious indoor air pollution and bring a lot of health benefit in cold season when people use coal for heating.

In conclusion, the current version is more like a technical paper and the contents and organization need to be improved a lot to meet the standard of Nature Communications.

Reviewer #2 (Remarks to the Author):

The paper evaluates the carbon impact of an important air pollution policy in China, namely, replacing raw coal rural heating with electrified heating. The implications of adding a significant amount of electric load onto the coal-intensive northern China grid is an important area of study. It is a timely piece, but which could benefit from clearer documentation and a refined time dimension in the model.

First, according to Figure 2b, it appears that coal is always on the margin, only varying in efficiency. However, given curtailment, wouldn't we expect to see some periods in the day in which renewables are on the margin? More conventionally, a marginal CO₂ intensity of the grid is used, in which case there could be periods of 0 CO₂ intensity.

Second, the paper ignores the distribution infrastructure necessary to meet these increased loads. It proposes using 8-kW EH or HP (line 838) as representative technologies, which is presumably many times current peak power consumption per household. Is this feasible given the level of the grid? The household cost analysis is very instructive, but there may be other socialized costs to consider.

Relatedly, large rooftop solar systems that must put power back on the grid: what amount of curtailment in the distribution infrastructure might we expect?

More fundamentally, I am concerned by statements like "7.79-12.12 kW rooftop solar system to totally offset the carbon emission increase" (530-1), which appears to assume that the solar is displacing a unit of equal carbon intensity as the marginal unit when the heating system is used. This seems unlikely even without altering commitment patterns; and when considering commitment changes, even less true. Fig 4c presumably makes the same assumption.

A similar argument appears to be made here: "To totally offset the carbon emission increase, we

discover that the requirement for additional renewable energy is 23.34 TWh, accounting for 1.64% of the total renewable generation in 2015...Note that consuming an additional 2.0% of the national renewable energy could reduce 74.04% of wind and solar curtailment in Northern China." (374-382). This makes the very large assumption that heating loads are all coincident--or could be made to be coincident--with existing curtailment patterns.

I believe the paper can address--but currently does not--a broader framing that has appeared in many articles in recent years on China's climate and environmental policies: the question of co-benefits. That is, to what extent do these policies generate co-benefits or disbenefits for other environmental goals? The rural heating example could fit nicely into this literature, which mostly focuses on macro-level energy systems.

A few notes on sources:

Citation link (9) is broken. These figures presumably relate to ambient pollution, while primary coal burning in households can also contribute to indoor air pollution. Is there a source on the latter?

Citation (14) on peak daily loads indicates this was the largest daily consumption in winter, not peak load (e.g., hourly). It is not clear that increased heating load would have been coincident with peak load.

On lines 603-4, "Based on real-world datasets, we collect the parameters of thermal generators...":

What is the source of these datasets?

Regarding the heating data, is the outdoor air temperature a daily average? The method seems sound on average, but for connections to power systems, an hourly profile is advised (see above).

What is the time resolution of the EnergyPlus heating data requirements? If it is relatively coarse, are there other sources to further identify hourly profiles?

A few notes on terminology:

The whole paper could benefit from a consistent and commonly used terminology to distinguish energy consumption (e.g., over the day or year) and power (e.g., peak load).

Seeking further clarification:

Raw coal and thermal coal appear to be directly compared. However, I expect that these will have different qualities (also perhaps regional differences).

How are households converted to populations?

Regarding prices, is there any worry that large heating loads may push consumers into a higher tier tariff?

Some basic details on the survey (lines 155-6) are important--how it was conducted, by whom, representativeness, etc.

Reply to the comments

Manuscript: NCOMMS-19-39588-T

Authors: Jianxiao Wang, Haiwang Zhong, Zhifang Yang, Mu Wang, Daniel M Kammen, Zhu Liu, Ziming Ma, Qing Xia, Chongqing Kang

Title: Exploring the Trade-offs between Electric Heating Policy and Carbon Mitigation in China

Dear Reviewers:

We would like to thank the Reviewers for the insightful comments and for the thorough revision of the manuscript. We have made great efforts to extend the results to environment field and provide more compelling justifications. All the comments and suggestions have been carefully addressed.

Reviewers' comments:

Reviewer #1 (Remarks to the Author):

Wang et al. performed an analysis of the impacts of China's Electric Heating Policy (EHP) on the carbon dioxide and air pollutant emissions. This is a typically multi-disciplinary study relating energy (e.g., electricity, coal and renewable energy), environment (e.g., carbon and air pollutant) and economy (e.g., cost). It looks that the authors are familiar with the energy system. But there are some problems when they extend the results to the field of environment. For example, the authors use the same CO₂ emission factor of coal used in power plant and residential heating. But, the coal combustion of power plant is generally more sufficient (i.e., the oxidation rate is higher), so the carbon emission level of 1kg coal in power plant is higher than that of residential heating. Another example is NO₂, which comes from the reaction of N₂ and O₂ in the atmosphere at high temperature (content of N in fossil fuel is very low). The authors didn't seem to realize these problems.

Answer:

We appreciate the Reviewer's invaluable suggestions and comments.

As noted by the Reviewer, the coal combustion of power plants is generally more sufficient than that of rural residential heating. Therefore, the carbon emission factor of thermal coal is higher than that of rural raw coal. In the revised manuscript, the emission factors of raw coal and thermal coal have been corrected as follows.

According to ref. [1], the heating value of standard coal is $h^{Coal}=2.93\times 10^7$ J/kg, and the net carbon content per energy is $\alpha^{Coal}=26.59$ tC/TJ. The difference between raw coal and thermal coal lies in the oxidization rate. The oxidization rate of raw coal is set as $o^{HCoal}=83.7\%$, and that of thermal coal is set as $o^{GCoal}=99\%$ [1]. Therefore, the emission factors of raw coal and thermal coal, denoted by e^{HCoal} and e^{GCoal} , are calculated below:

$$e^{HCoal} = o^{HCoal} \times h^{Coal} \times \alpha^{Coal} = 2.39 \text{ tCO}_2/\text{t} \quad (1)$$

$$e^{GCoal} = o^{GCoal} \times h^{Coal} \times \alpha^{Coal} = 2.83 \text{ tCO}_2/\text{t} \quad (2)$$

All the simulation results have been accordingly revised.

In addition, the evaluation of pollutant emissions has been removed from the revised manuscript because this paper is aimed at exploring the link between China's Electric Heating Policy (EHP) and national carbon mitigation. We mainly focus on developing a theoretical model to quantify the coal consumption for space heating from electric power and rural resident sectors. Therefore, it is beyond the core scope of this paper regarding how to accurately determine the pollutant emission factors, which deserves an in-depth investigation in our future work.

[1] Liu, Z. *et al.* Reduced carbon emission estimates from fossil fuel combustion and cement production in China. *Nature* **524**, 335-338 (2015).

The authors conclude that the EHP will significantly increase the CO₂ emissions but reduce air pollutant emissions (e.g., SO₂). I am not surprised about this conclusion because the EHP policy was designed to mitigate the air pollution in China. Especially in China's rural area, the EHP could significantly reduce the serious indoor air pollution and bring a lot of health benefit in cold season when people use coal for heating.

Answer:

We appreciate the Reviewer's invaluable suggestions and comments.

The authors agree with the reviewer that China's Electric Heating Policy was designed to mitigate air pollution and improve air quality. However, the objective of this research is to investigate China's Electric Heating Policy and the disbenefits on other environmental goals. The major contribution is to develop a theoretical model enabling **a refined quantification** of hourly electric heating load and associated carbon emissions from both electric power and rural resident sectors. In contrast to existing literature, this paper is **the first attempt** to provide such a tool and platform to achieve high-resolution simulation. Taking advantage of the proposed model, we can quantitatively evaluate the impacts of various critical factors under different boundary conditions **instead of highly relying on macro statistical data**.

To this end, we further highlight the motivation and contribution of this paper:

1) We propose an integrated theoretical model to quantify the emissions induced by China's EHP considering both generation and rural residential heating sectors. The thermal coal consumption and emission of power grids are estimated by using **an hourly-scheduling unit commitment model**, which is formulated as a mixed integer linear programming (MILP) problem. The raw coal for space heating of a single household is simulated by using **EnergyPlus**, which is a software developed by National Renewable Energy Laboratory (NREL), Lawrence Berkeley National Laboratory (LBNL), etc. and sponsored by the Department of Energy (DOE).

2) Based on the proposed model, we quantify the extent that China's EHP can contribute to national carbon emissions. We find that in 2015, an additional 101.69-162.89 megatons of CO₂ could potentially be emitted if EHP was implemented in 45%-55% of rural residents in Northern China. **Such incremental carbon emission approximately accounts for 31.02%-49.69% of France's annual emission**. With the development of EHP, we estimate that the carbon emissions will keep discharging in the following years. In 2020, the incremental carbon emission is expected to reach 130.03-197.87 megatons. However, the growth of carbon

emission will slow down due to China’s urbanization progress. In 2030, the carbon emission increase induced by EHP will drop to 119.19-177.47 megatons.

3) To address the trade-offs between EHP and national carbon mitigation, two practical pathways are proposed for China and other countries with similar situations. The first is to balance electric heating load with renewable energy. Specifically, we discuss the impacts of improving the **accommodation of provincial renewable generation** and installing **distributed energy resources**. The second is to improve the **efficiency of electric heating** in order to directly limit electric load increase. Furthermore, we analyze the average **annualized cost per household in 25 cities in Northern China** to evaluate the feasibility of different electric heating technologies.

Hopefully, our proposed integrated model and strategy will reduce pollutant emissions and accelerate the decarbonization of the global electricity sector.

In addition, as noted by the Reviewer, especially in China's rural area, the EHP could significantly reduce the serious indoor air pollution and bring a lot of health benefit in cold season when people use coal for heating. This is really an interesting and important topic, which might be our future work.

In conclusion, the current version is more like a technical paper and the contents and organization need to be improved a lot to meet the standard of Nature Communications.

Answer:

We appreciate the Reviewer’s invaluable suggestions and comments.

The writing and structure of this paper have been significantly improved to meet the standard of Nature Communications. The paper structure is organized as follows:

The paper mainly consists of three sections. In the introduction, we explain the background and motivation of this paper, i.e., to what extent does China’s EHP contribute to national carbon emissions thus impeding carbon mitigation footprint?

In the section of results, we elaborate the simulation and findings about provincial carbon

emissions, national carbon emissions and techno-economic analysis. For provincial carbon emissions, the simulations based on Hebei, Henan, Shandong and Shanxi are conducted, considering the impacts of the uncertainties in policy implementation rate and electric heating mix. Additionally, the key factors leading to provincial diversity are analyzed. For national carbon emissions, we extend the results in four provinces to Northern China, and forecast the national carbon emissions caused by EHP in 2020 and 2030. For techno-economic analysis, two low-carbon electric heating pathways are proposed, including developing renewable energy and improving electric heating efficiency. Cost analysis is carried out for different electric heating pathways in 25 cities in Northern China.

In the section of discussions, we summarize three policy suggestions for China and other developing countries with similar situations.

Reviewer #2 (Remarks to the Author):

The paper evaluates the carbon impact of an important air pollution policy in China, namely, replacing raw coal rural heating with electrified heating. The implications of adding a significant amount of electric load onto the coal-intensive northern China grid is an important area of study. It is a timely piece, but which could benefit from clearer documentation and a refined time dimension in the model.

Answer:

We appreciate the Reviewer's invaluable suggestions and comments.

The manuscript has been carefully revised according to the Reviewer's comments. The detailed replies are presented one by one as follows.

First, according to Figure 2b, it appears that coal is always on the margin, only varying in efficiency. However, given curtailment, wouldn't we expect to see some periods in the day in which renewables are on the margin? More conventionally, a marginal CO₂ intensity of the grid is used, in which case there could be periods of 0 CO₂ intensity.

Answer:

We appreciate the Reviewer's invaluable suggestions and comments.

As noted by the Reviewer, coal-fired units are on the margin in Fig. 2b. This is because there is no renewable energy curtailment in the base case. As the power network constraints are ignored in this paper, zero carbon-intensity renewable energy will be fully accommodated since the system-wide electric load is greater than renewable power. The remaining loads are balanced by coal-fired generators. Therefore, it appears that coal-fired units are always on the margin.

However, as shown in Fig.r 1 (Fig.4a in the revised manuscript), a higher level of renewable energy penetration yields an increase in renewable energy curtailment, indicating that renewable energy can be marginal generation resources in some cases.

Fig.r 1 Carbon emissions in the four provinces with the integration of renewable energy. The radius of each bubble represents the curtailment rate for additional renewable energy, and the center indicates the incremental carbon emission caused by EHP.

Second, the paper ignores the distribution infrastructure necessary to meet these increased loads. It proposes using 8-kW EH or HP (line 838) as representative technologies, which is presumably many times current peak power consumption per household. Is this feasible given the level of the grid? The household cost analysis is very instructive, but there may be other socialized costs to consider.

Answer:

We appreciate the Reviewer’s invaluable suggestions and comments.

Chinese government invested 4 trillion Yuan answering the 2008 financial crisis, a large fraction of which has been used for infrastructure construction including the transformation of rural power grid. As a result, the distribution infrastructures in many rural areas have been upgraded and expanded. This fact lays the foundation for the implementation of Electric Heating Policy. In this paper, it is assumed that the distribution grid capacity is sufficient.

As noted by the Reviewer, there may be other socialized costs in addition to the household cost. However, the distribution infrastructures in some regions in China have already been upgraded. Moreover, the expansion of distribution networks is generally launched and managed by Chinese government and power grid companies. The allocation of socialized costs is a complicated mechanism, which deserves a systematic study in our future work. Therefore, the costs associated with the distribution infrastructure necessary to meet the electric heating loads are not considered in this paper. Instead, we focus on the investment and operation costs induced by the heating devices of rural residents.

Relatedly, large rooftop solar systems that must put power back on the grid: what amount of curtailment in the distribution infrastructure might we expect?

Answer:

We appreciate the Reviewer’s invaluable suggestions and comments.

Admittedly, large rooftop solar systems can put power back to the connected grid. Thus, Fig. 4b has been added in the revised manuscript, shown as follows.

Fig.r 2 Carbon emission reduction capability of distributed PV in the four provinces. The lines represent the carbon emission reduction per capita per kW, and the bars show the curtailment of solar energy.

The incremental carbon emissions caused by EHP can be effectively limited by installing distributed photovoltaic (PV) resources. However, the accommodation capability for PV declines, leading to more solar energy curtailment and thus a decreasing carbon emission reduction rate. Additionally, the curtailment curves in a typical week of four provinces are presented in Fig.r 3.

Fig.r 3 Solar power curtailment during a week in the four provinces with different levels of household PV capacity.

More fundamentally, I am concerned by statements like "7.79-12.12 kW rooftop solar system to totally offset the carbon emission increase" (530-1), which appears to assume that the solar is displacing a unit of equal carbon intensity as the marginal unit when the heating system is used. This seems unlikely even without altering commitment patterns; and when considering commitment changes, even less true. Fig 4c presumably makes the same assumption.

Answer:

We appreciate the Reviewer’s invaluable suggestions and comments.

In fact, we are NOT assuming that the solar is displacing a unit of equal carbon intensity as the marginal unit. The results are obtained by using the proposed hourly unit commitment (UC) model. On the premise of a given installed PV capacity, we input the boundary conditions into the UC model, and the carbon emissions from the power generation sector can be simulated.

As can be observed from Fig. 4c in the original manuscript, the relationship between provincial carbon emission and household PV capacity is nonlinear. This is because unit commitment is used for simulation rather than simply replacing marginal unit with solar energy.

Fig.r 4 The relationship between carbon emission increase and a single household’s solar installed capacity.

It should be noted that our proposed theoretical model is the first attempt to provide a tool

and platform to enable a refined quantification of hourly electric heating load and associated carbon emissions from both electric power and rural resident sectors. Taking advantage of the proposed model, we can quantitatively evaluate the impacts of various critical factors under different boundary conditions instead of highly relying on statistical data.

A similar argument appears to be made here: "To totally offset the carbon emission increase, we discover that the requirement for additional renewable energy is 23.34 TWh, accounting for 1.64% of the total renewable generation in 2015...Note that consuming an additional 2.0% of the national renewable energy could reduce 74.04% of wind and solar curtailment in Northern China." (374-382). This makes the very large assumption that heating loads are all coincident--or could be made to be coincident--with existing curtailment patterns.

Answer:

We appreciate the Reviewer's invaluable suggestions and comments.

In this paper, the heating loads are NOT assumed to be coincident with existing renewable energy curtailment patterns. The hourly profiles of provincial renewable energy are collected from historical datasets. Then we scale up the renewable power profiles and recalculate the UC model. Given a scenario with updated renewable power and electric heating load, we reset the boundary conditions of UC, and obtain the associated carbon emissions and renewable energy curtailment by optimizing UC.

For example, with an additional 1% renewable energy, we multiply the original renewable power profile by 1.01. Based on these data, the unit commitment model is optimized, and the total thermal coal consumption as well as renewable energy curtailment for this scenario can be obtained.

I believe the paper can address--but currently does not--a broader framing that has appeared in many articles in recent years on China's climate and environmental policies: the question of co-benefits. That is, to what extent do these policies generate co-benefits or disbenefits for other environmental goals? The rural heating example could fit nicely into this literature, which mostly focuses on macro-level energy systems.

Answer:

We appreciate the Reviewer's invaluable suggestions and comments.

As the reviewer pointed out, this paper is aimed at exploring the incompatibility between China's Electric Heating Policy and national carbon mitigation. To reduce pollutant emissions and improve air quality, China has enacted a series of policies since 2015 to substitute electricity for in-home combustion for rural residential heating. "Plan for Winter Clean Heating in Northern China (2017-2021)", issued by National Energy Administration in 2017, enforced strict regulations to switch raw coal to electric heating among rural residents in Beijing, Tianjin and other 14 provinces in Northern China. The Electric Heating Policy (EHP) has contributed to significant improvements in indoor air quality, benefiting hundreds of millions of people. However, this shift has resulted in a dramatic increase in carbon emissions from China's electricity sector, yielding a great impact on global carbon mitigation.

To further highlight the discovery of this paper, we summarize in the revised manuscript that the government must explore the potential incompatibility between any new policy and the existing ones. According to our analyses, the underestimation of the greenhouse effect caused

by EHP can impede China's carbon mitigation process in the future. Meanwhile, an increasing penetration of electric heating may lead to the shortage of generation capacity and flexible load-following resources, thus threatening the secure and reliable operation of power grids.

In our future work, the co-benefits or disbenefits for other environmental goals deserve a systematic study.

A few notes on sources:

Citation link (9) is broken. These figures presumably relate to ambient pollution, while primary coal burning in households can also contribute to indoor air pollution. Is there a source on the latter?

Answer:

We appreciate the Reviewer's invaluable suggestions and comments.

Citation link (9) has been updated:

<http://energy.ingold.org/c/2013-06-07/c1880147.html>

Admittedly, substituting electric heating in place of raw coal contributes to the improvement of indoor air and human health. However, we mainly focus on the environmental impacts of electric heating on macro-energy systems, and we have not found any data source regarding indoor air pollution. This topic deserves an in-depth investigation in the future.

Citation (14) on peak daily loads indicates this was the largest daily consumption in winter, not peak load (e.g., hourly). It is not clear that increased heating load would have been coincident with peak load.

Answer:

We appreciate the Reviewer's invaluable suggestions and comments.

As the Reviewer noted, citation (14) refers to the largest daily load consumption rather than peak hourly load. Therefore, in the revised manuscript, it has been revised as: "Compared with 2017, the largest daily electricity consumption in January 2018 increased by over 15%".

On lines 603-4, "Based on real-world datasets, we collect the parameters of thermal generators...":

What is the source of these datasets?

Answer:

We appreciate the Reviewer's invaluable suggestions and comments.

These datasets come from two sources:

1) Some of the data such as the installed capacity of thermal generators are collected from the reports published by research institutes in China, e.g., "China Energy & Electricity Outlook" by State Grid Energy Research Institute and "Development Report of China's Electric Power Industry" by China Electricity Council.

2) Other data are originated from the project cooperation with power grid companies in the four provinces, including thermal coal consumption rate, hourly load and renewable profiles, etc.

Regarding the heating data, is the outdoor air temperature a daily average? The method seems sound on average, but for connections to power systems, an hourly profile is advised (see above).

Answer:

We appreciate the Reviewer’s invaluable suggestions and comments.

In this paper, the outdoor air temperature is NOT a daily average but an hourly value, which is provided by the following link:

<https://www.energyplus.net/weather>

To systematically evaluate the heating coal and load consumption in a province, we select the outdoor air temperature in several cities in each province. In HB, we use the weather data from three cities, i.e., Raoyang, Shijiazhuang and Xingtai. In HN, we use the weather data from seven cities, including Anyang, Lushi, Nanyang, Shangqiu, Xinyang, Zhengzhou and Zhumadian. In SD, we use the weather data from eight cities, including Chaoyang, Chengshantou, Huimin, Jinan, Juxian, Longkou, Weifang and Yanzhou. In SX, we use the weather data from seven cities, including Datong, Houma, Jiexiu, Taiyuan, Yuanping, Yuncheng and Yushe.

Fig.r 5 shows an average hourly outdoor air temperature during one day in HB, HN, SD and SX.

Fig.r 5 Average hourly outdoor air temperature during one day in HB, HN, SD and SX.

What is the time resolution of the EnergyPlus heating data requirements? If it is relatively coarse, are there other sources to further identify hourly profiles?

Answer:

We appreciate the Reviewer’s comments.

The time resolution of the electric heating load data provided by EnergyPlus is hourly. Given different sizes of households, Fig.r 6 shows an average hourly electric heating load during one day in HB, HN, SD and SX.

(50,70] m²

(70,90] m²

(90,120] m²

(120,150] m²

Fig.r 6 Average hourly electric heating load during one day in HB, HN, SD and SX.

A few notes on terminology:

The whole paper could benefit from a consistent and commonly used terminology to distinguish energy consumption (e.g., over the day or year) and power (e.g., peak load).

Answer:

We appreciate the Reviewer's invaluable suggestions and comments.

In the revised manuscript, the concepts of energy consumption and power have been clarified. For example, it has been revised as: "Compared with 2017, the largest daily electricity consumption in January 2018 increased by over 15%".

Seeking further clarification:

Raw coal and thermal coal appear to be directly compared. However, I expect that these will have different qualities (also perhaps regional differences).

Answer:

We appreciate the Reviewer's invaluable suggestions and comments.

Firstly, the coal combustion of power plants is generally more sufficient than that of rural residential heating. Therefore, the carbon emission factor of thermal coal is higher than that of rural raw coal. In the revised manuscript, the emission factors of raw coal and thermal coal have been corrected as follows.

According to ref. [2], the heating value of standard coal is $h^{Coal}=2.93\times 10^7$ J/kg, and the net carbon content per energy is $\alpha^{Coal}=26.59$ tC/TJ. The difference between raw coal and thermal coal lies in the oxidization rate. The oxidization rate of raw coal is set as $o^{HCoal}=83.7\%$, and that of thermal coal is set as $o^{GCoal}=99\%$ [2]. Therefore, the emission factors of raw coal and thermal coal, denoted by e^{HCoal} and e^{GCoal} , are calculated below:

$$e^{HCoal} = o^{HCoal} \times h^{Coal} \times \alpha^{Coal} = 2.39 \text{ tCO}_2/\text{t} \quad (3)$$

$$e^{G_{\text{Coal}}} = o^{G_{\text{Coal}}} \times h^{\text{Coal}} \times \alpha^{\text{Coal}} = 2.83 \text{tCO}_2/\text{t} \quad (4)$$

Secondly, we have considered the regional difference of thermal coal in this paper. When estimating the carbon emissions in each province in Northern China, the provincial average thermal coal consumption rates in 2015 are distinguished, shown in Fig.r 7.

Fig.r 7 Provincial average thermal coal consumption rates in 2015.

[2] Liu, Z. *et al.* Reduced carbon emission estimates from fossil fuel combustion and cement production in China. *Nature* **524**, 335-338 (2015).

How are households converted to populations?

Answer:

We appreciate the Reviewer's invaluable suggestions and comments.

The households are converted to populations as follows:

The households' data are simulated by EnergyPlus and the populations are obtained from the Sixth National Census in China.

According to the Sixth National Census in China, the 2015 rural populations in HB, HN, SD and SX were 36.14, 50.39, 42.33 and 16.48 million, respectively. The 2015 rural population in 16 provinces (autonomous regions and municipalities) in Northern China was 256.19 million. According to the population target planning in 2020, the rural populations in the HB, HN, SD and SX will be 33.00, 49.06, 35.88 and 16.60 million, respectively. In 2030, the rural populations in the four provinces are expected to reach 25.74, 39.10, 26.67 and 9.67 million, respectively.

To systematically evaluate the households' heating energy consumption considering different sizes, we conduct sensitivity analyses on the sizes of houses. We scan the length, width and height of houses from 5 m to 20 m, from 3 m to 12 m, and from 3 m to 5 m, respectively.

The housing area data are collected from “Report on Chinese Residential Energy Consumption”, published by the National Academy of Development and Strategy, Renmin University of China. The housing areas are divided into eight intervals, i.e., [15,30], (30,50], (50,70], (70,90], (90,120], (120,150], (150,180] and (180,250] m², accounting for 1.05%, 3.48%, 7.67%, 13.24%, 24.39%, 16.72%, 14.29% and 19.16%, respectively. Let \bar{Q}_j^{HCoal} and $\bar{P}_j^{Elec}(t)$ be the average heating coal consumption and hourly electric heating load at time slot t of the households in the j^{th} area interval, respectively. The average household heating coal consumption \bar{Q}^{HCoal} and the average household hourly electric heating load $\bar{P}^{Elec}(t)$ in a province can be calculated as follows:

$$\bar{Q}^{HCoal} = \sum_{j=1}^8 \gamma_j \bar{Q}_j^{HCoal} \quad (5)$$

$$\bar{P}^{Elec}(t) = \sum_{j=1}^8 \gamma_j \bar{P}_j^{Elec}(t) \quad (6)$$

where $\gamma_j, j = 1, 2 \dots 8$ represents the proportion of the j^{th} area interval, i.e., 1.05%, 3.48%, 7.67%, 13.24%, 24.39%, 16.72%, 14.29% and 19.16%, respectively. Then a province’s total heating coal consumption Q^{HCoal} and total hourly electric heating load $P^{Elec}(t)$ can be obtained by using the following equations:

$$Q^{HCoal} = \bar{Q}^{HCoal} \times \rho \times PIR \quad (7)$$

$$P^{Elec}(t) = \bar{P}^{Elec}(t) \times \rho \times PIR \quad (8)$$

where ρ is provincial rural resident population; and PIR represents the policy implementation rate.

Regarding prices, is there any worry that large heating loads may push consumers into a higher tier tariff?

Answer:

We appreciate the Reviewer’s invaluable suggestions and comments.

Regarding prices, there are various policies in different areas to support the implementation of Electric Heating Policy. For example, the rural residents who substitute electric heating in place of raw coal can enjoy the lowest tier tariff in heating season in SD [3]. The rural residents in SX can select the flat rate for electric heating instead of time-of-use tariff and tier tariff [4]. Currently, the rural residents participating in EHP can enjoy relatively low electricity prices supported by Chinese government. As a result, large heating loads may not push consumers into a higher tier tariff, and thus the impact of heating loads on tariffs is not considered in this paper.

[3] Good news! "One household, one meter" electric heating residents in SD province implement the lowest tier tariff.

<http://sd.people.com.cn/GB/n2/2018/1125/c166192-32328061.html>

[4] Development and Reform Commission of Shanxi province issues electric heating price

policy.

<http://www.china-heating.com/news/2017/38654.html>

Some basic details on the survey (lines 155-6) are important--how it was conducted, by whom, representativeness, etc.

Answer:

We appreciate the Reviewer's invaluable suggestions and comments.

We have contacted the government in Hebei province since Hebei is a pioneering province in Northern China implementing EHP. HB government set a goal in 2018 for the annual development of various electric heating devices in different cities, shown as follows:

TABLE 1 Annual development of various electric heating devices in Hebei in 2018
(Unit: Household)

City	County	Electric heater	Photovoltaic	Others
Shijiazhuang	Wuji	14258	0	0
	Jinzhou	8560	0	0
	Zhaoxian	9021	0	0
	Xinle	4624	0	0
	Xingtang	3663	0	0
	Gaoyi	8293	0	0
	Shenze	4611	0	0
	Yuanshi	6213	0	0
	Lingshou	4906	0	0
	Pingshan	5323	0	0
	Zanhuang	4161	0	0
	Luancheng	0	315	0
	Others	0	0	5000
	Total	73633	315	5000
Zhangjiakou	Chongli	829	208	0
	Zhangbei	876	260	0
	Huailai	1000	0	0
	Xuanhua	800	0	0
	Zhuolu	800	0	0
	Guyuan	500	0	0
	Huaian	406	0	0
	Kangbao	400	0	0
	Shangyi	398	0	0
	Yuxian	322	0	0
	Qiaoxi	302	0	0
	Jingkai	300	0	0
	Qiaodong	236	0	0
	Chicheng	173	0	0
	Yangyuan	150	0	0
Others	0	0	1495	
	Total	7492	468	1495
Chengde	Shuangqiao	0	170	0
	Pingquan	300	0	0
	Chengde	180	0	0
	Shuangluan	200	160	0
	Fengning	200	0	0
	Longhua	100	0	0

	High-tech Zone	100	0	0
	Kuancheng	100	0	0
	Yingzi	10	0	0
	Others	0	0	0
	Total	1190	330	0
Qinhuangdao	Qinglong	0	216	0
	Haigang	0	165	0
	Changli	0	158	0
	Shanhaiguan	0	0	50
	Lulong	0	0	26
	Others	0	0	0
	Total	0	539	76
Tangshan	Yutian	3622	0	0
	Fengnan	2826	0	0
	Qian'an	3178	0	0
	Luannan	3728	245	0
	Laoting	2695	0	0
	Luanxian	2355	0	0
	Fengrun	3533	0	0
	Zunhua	6862	0	0
	Qianxi	2831	0	0
	Caofeidian	2837	0	0
	Guye	2000	0	0
	Others	0	0	0
Total	36467	245	0	
Baoding	Qingyuan	6197	0	0
	Wangdu	2150	0	0
	Anguo	6489	0	0
	Gaobeidian	1049	0	0
	Lixian	1754	0	0
	Quyang	2597	0	0
	Shunping	4846	0	0
	Boye	3511	0	300
	Fuping	0	208	0
	Others	0	0	0
Total	28593	208	300	
Cangzhou	Suning	2004	0	0
	Qingxian	3000	0	0
	Xianxian	140	100	0
	Nanpi	100	0	0
	Xinhua	5682	0	0
	Wuqiao	793	0	0
	Huanghua	470	0	0
	Haixing	1000	0	0
	Botou	620	0	0
	Dongguang	150	0	0
	Yunhe	261	0	0
	Others	0	0	0
Total	14220	100	0	
Hengshui	Zaoqiang	0	630	0
	Wuyi	2992	0	0
	Shenzhou	1673	0	0
	Gucheng	1413	0	0

	Wuqiang	840	0	0
	Raoyang	1187	0	0
	Jingxian	1732	0	0
	Fucheng	1975	0	0
	Binhu	27	0	0
	Others	0	0	0
	Total	11839	630	0
Xingtai	Nanhe	7120	0	0
	Ningjin	5750	0	0
	Renxian	6209	120	100
	Weixian	6218	150	5000
	Longyao	2804	0	0
	Shahe	7917	0	0
	Baixiang	5248	120	300
	Nangong	5576	0	100
	Xingtai	4969	0	0
	Neiqiu	5465	0	100
	Qinghe	4081	166	0
	Development Zone	2568	0	0
	Guangzong	5234	0	0
	Pingxiang	3858	0	0
	Lincheng	1204	243	0
	Linxi	3396	0	0
	Qiaoxi	864	0	0
	Julu	1805	0	0
	Xinhe	578	240	960
	Others	0	0	0
Total	80864	1039	6560	
Handan	Cixian	7132	196	15
	Linzhang	2798	100	1015
	Jinan New Area	1774	0	0
	Cheng'an	3248	110	15
	Fengfeng	4063	145	5015
	Feixiang	0	420	1315
	Wuan	2000	100	2015
	Weixian	4318	300	15
	Daming	2457	400	15
	Guantao	3208	550	15
	Quzhou	5000	100	15
	Hanshan	5672	0	0
	Congtai	4745	0	0
	Jize	3000	100	15
	Qiuxian	0	400	15
	Fuxing	2965	0	0
	Shexian	0	100	15
	Guangping	1000	130	15
	Others	0	0	0
	Total	53380	3151	9510
Xiong'an	5000	0	0	
Langfang	82	0	0	
Dingzhou	0	850	0	
Xinji	6000	583	100	
Total	318760	8458	23041	

The data in TABLE 1 are adopted in this paper. The total numbers of rural households used electric heaters (EHs), photovoltaic-powered electric heating (PVEH) and other devices are 318760, 8458 and 23041, respectively. Note that other devices include heat pumps (HPs), geothermal energy, graphene electric heating, etc., which have different heating efficiencies. Due to data limitation, we use HPs as a representative technology in this paper, and systematically evaluate the impacts of heating efficiency by scanning the coefficient of performance (COP) from 250% to 400%.

According to the total numbers in TABLE 1, the average proportions of EHs, PVEH and HPs are set as 91.01%, 2.41% and 6.58%, respectively. Additionally, we have considered the uncertainties in the proportion of electric heating mix and policy implementation rate in this paper.

In the end, the authors would like to thank the Reviewers for all the invaluable suggestions and comments on this paper. It is your kind help that makes our work better.

REVIEWER COMMENTS

Reviewer #1 (Remarks to the Author):

The paper is much improved after revision. The authors made great effort to improve the quality of data, figures/tables and results. I think it is acceptable as its current version.

However, it seems the authors cannot include the air pollutant emissions at this point, although all comments have been addressed. I suggest to delete related contents about air pollution, and focus on the CO₂ emissions in the introduction section.

This study includes several models and a lot of data and assumptions. I have also concerns about the uncertainty in the simulations. The author should add some contents to clearly show the uncertainties/limitations in their conclusions before discussion.

Reviewer #2 (Remarks to the Author):

The authors have addressed the substantive aspects of each of my comments satisfactorily.

However, I believe some additional clarification is necessary in the text regarding the network representation of the UC model. As the authors note in their rebuttal:

"As the power network constraints are ignored in this paper, zero carbon-intensity renewable energy will be fully accommodated since the system-wide electric load is greater than renewable power."

Indeed, this is an important caveat. The current model cannot replicate current curtailment rates because of this simplification, hence it will underestimate future curtailment due to network congestion. While the formulation in "Electricity dispatch model" (l. 514) does not claim incorporation of networks, this modeling simplification should be noted in the discussion as a limitation of the modeling results. Both transmission networks as well as distribution networks (notwithstanding the authors' response to the question on distribution grid capacity, which would nevertheless require further study) should be included as additional potential sources of curtailment.

Reply to the comments

Manuscript: NCOMMS-19-39588A

Authors: Jianxiao Wang, Haiwang Zhong, Zhifang Yang, Mu Wang, Daniel M Kammen, Zhu Liu, Ziming Ma, Qing Xia, Chongqing Kang

Title: Exploring the Trade-offs between Electric Heating Policy and Carbon Mitigation in China

Dear Reviewers:

We would like to thank the Reviewers for the insightful comments and for the thorough review of the manuscript. We have carefully revised the manuscript to clearly show the impacts of various uncertainties and explain the limitations of electricity dispatch model. The comments and suggestions have been addressed point by point.

REVIEWER COMMENTS

Reviewer #1 (Remarks to the Author):

The paper is much improved after revision. The authors made great effort to improve the quality of data, figures/tables and results. I think it is acceptable as its current version.

However, it seems the authors cannot include the air pollutant emissions at this point, although all comments have been addressed. I suggest to delete related contents about air pollution, and focus on the CO₂ emissions in the introduction section.

Answer:

We appreciate the Reviewer's invaluable suggestions and comments.

The introduction section has been carefully revised to simplify the contents related to air pollution, but focus on carbon emissions instead. It should be noted that some contents about air pollution are reserved because China's Electric Heating Policy is designed for reducing the carbon and pollutant emissions from rural residents, and we think it is necessary to briefly introduce this background.

This study includes several models and a lot of data and assumptions. I have also concerns about the uncertainty in the simulations. The author should add some contents to clearly show the uncertainties/limitations in their conclusions before discussion.

Answer:

We appreciate the Reviewer's invaluable suggestions and comments.

As noted by the Reviewer, this study includes several models and a lot of data, which are summarized as follows for a better explanation:

TABLE R1 Required data in rural heating assessment model

No.	Title	Source
1	Rural resident population	Sixth national census in China
2	Housing area	Report on Chinese residential energy consumption

3	Weather	World Meteorological Organization
4	Indoor comfort temperature	Indoor air quality standard GB/T18883-2002
5	Rooftop solar power	National Renewable Energy Laboratory
6	Coal oxidization rate	Ref. [r1]
7	Policy implementation rate	Plan for winter clean heating in Northern China (2017-2021)
8	Electric heating mix	Survey data from Hebei government

TABLE R2 Required data in electricity dispatch model

No.	Title	Source
1	Thermal generators	State Grid Corporation of China and Development Report of China's Electric Power Industry
2	Renewable power	State Grid Corporation of China and Development Report of China's Electric Power Industry
3	Electric power load	State Grid Corporation of China and China Energy & Electricity Outlook
4	Tieline power	State Grid Corporation of China and China Energy & Electricity Outlook
5	Coal oxidization rate	Ref. [r1]

[r1]. Liu, Z. *et al.* Reduced carbon emission estimates from fossil fuel combustion and cement production in China. *Nature* **524**, 335-338 (2015).

Admittedly, uncertainties are very important in the simulations. Therefore, we focus on two major sources of uncertainties in TABLE R1, i.e., the policy implementation rate (PIR) and the electric heating mix (EHM).

PIR refers to the population of rural residents using electric heating over that of provincial rural residents. According to the “Plan for Winter Clean Heating in Northern China (2017-2021)”, 50% of rural residents in Beijing, Tianjin and 14 other provinces in Northern China must substitute electric heating for raw coal by 2019, and this PIR is required to reach 70% by 2021. However, PIR is a highly uncertain factor, depending on various events (such as the financial support from the government). Therefore, we investigate the impact of PIR uncertainty by scanning PIR within the range of $\pm 5\%$.

EHM refers to the proportions of different electric heating devices including electric heater (EH), heat pump (HP) and photovoltaic-powered electric heating (PVEH). According to the data in 11 cities in Northern China, EHs generally account for 79.24%-100% of the electric heating devices among rural residents. The average proportions of EHs, HPs and PVEH are 91.01%, 6.58% and 2.41%, respectively. Therefore, we investigate the impact of EHM uncertainty by scanning the proportions of EHs, HPs and PVEH within the intervals [80%, 100%], [0, 20%] and [0, 10%], respectively.

As illustrated in Fig.r1, the uncertainty in EHM (the boxplots) may yield a greater impact on the incremental carbon emissions than that in PIR (the bars). Based on the average EHM, the largest deviations of carbon emission induced by PIR uncertainty are 2.46, 3.76, 4.94 and 2.54 megatons in Hebei, Henan, Shandong and Shanxi, respectively. However, on the premise of a fixed PIR, the largest deviations of carbon emission caused by EHM uncertainty can reach 3.58, 6.22, 6.85 and 3.22 megatons in the four provinces, respectively.

Fig. r1 Carbon emission estimation in four provinces in Northern China after implementing EHP in 2015. Each three bars for a province represent the incremental carbon emissions with an average proportion of electric heating devices when the PIR equals 45%, 50% and 55%, respectively. The boxes represent the incremental carbon emissions acquired by scanning the proportions of EHs, HPs and PVEH.

Additionally, considering the joint uncertainty in PIR and EHM, the incremental carbon emission in Northern China in 2015 is estimated to vary from 101.69 to 162.89 megatons, as shown in Fig.r2 a. As illustrated in Fig.r2 b, we also consider the PIR and EHM uncertainty in the cases of 2020 and 2030.

Fig. r2 National impacts of China's EHP. a Comparisons of carbon emissions between China and other countries in 2015. The box with red dashed lines shows the variation of China's incremental carbon emission considering the uncertainty in the policy implementation rate and electric heating mix. **b** Incremental carbon emissions after implementing EHP in Northern China in 2015, 2020 and 2030. The intervals on the bars represent the variations of incremental carbon emissions. The radius of each bubble indicates the provincial population, and the center shows the provincial ruralization rate, i.e., the rural population over the total amount.

It is worth mentioning that based on our proposed theoretical model, the uncertainties in other data and parameters can be readily investigated.

Reviewer #2 (Remarks to the Author):

The authors have addressed the substantive aspects of each of my comments satisfactorily.

However, I believe some additional clarification is necessary in the text regarding the network representation of the UC model. As the authors note in their rebuttal:

"As the power network constraints are ignored in this paper, zero carbon-intensity renewable energy will be fully accommodated since the system-wide electric load is greater than renewable power."

Indeed, this is an important caveat. The current model cannot replicate current curtailment rates because of this simplification, hence it will underestimate future curtailment due to network congestion. While the formulation in "Electricity dispatch model" (l. 514) does not claim incorporation of networks, this modeling simplification should be noted in the discussion as a limitation of the modeling results. Both transmission networks as well as distribution networks (notwithstanding the authors' response to the question on distribution grid capacity, which would nevertheless require further study) should be included as additional potential sources of curtailment.

Answer:

We appreciate the Reviewer's invaluable suggestions and comments.

As pointed out by the Reviewer, power network constraints are not considered in the electricity dispatch model, which cannot exactly replicate the curtailment of renewable energy in China. Such simplification is made based on the following reasons.

On the one hand, recent years have witnessed China's great efforts to facilitate the accommodation of renewable energy. As declared by National Energy Administration, the total renewable power generation reached 2040 TWh in 2019, while the curtailed renewable energy was 51.5 TWh (Hydropower: 30 TWh, wind power: 16.9 TWh and solar power: 4.6 TWh). The national curtailment rate in 2019 was only 2.52%. Therefore, renewable energy curtailment may not become a significant impact factor.

On the other hand, we totally agree with the Reviewer that our proposed model will underestimate future curtailment of renewable energy due to the simplification for network congestion. In other words, the incorporation of network constraints will lead to more renewable curtailment, and thus a higher level of carbon emissions caused by Electric Heating Policy. Therefore, we claim that our proposed method and model provide a conservative estimation for national carbon emission increase. Also, we suggest that the Chinese government should pay more attention to the carbon reduction performance of developing inter-provincial renewable energy and distributed solar systems due to potential network congestion.

The manuscript has been carefully revised, and the discussions have been added.

In the end, the authors would like to thank the Reviewers for all the invaluable suggestions and comments on this paper. It is your kind help that makes our work better.

REVIEWERS' COMMENTS

Reviewer #1 (Remarks to the Author):

The authors have addressed my concerns. And the current version could be accepted for publication.

Response to Comments

Manuscript: NCOMMS-19-39588B

Authors: Jianxiao Wang, Haiwang Zhong, Zhifang Yang, Mu Wang, Daniel M Kammen, Zhu Liu, Ziming Ma, Qing Xia, Chongqing Kang

Title: Exploring the Trade-offs between Electric Heating Policy and Carbon Mitigation in China

Dear Reviewers:

We would like to thank the Reviewers for the insightful comments and for the thorough review of the manuscript.

Reviewer #1 (Remarks to the Author):

The authors have addressed my concerns. And the current version could be accepted for publication.

Answer:

We appreciate the Reviewer's encouragement.

In the end, the authors would like to thank the Reviewers for all the invaluable suggestions and comments on this paper. It is your kind help that makes our work better.